# Latent Representation Alignment for Offline Goal-Conditioned Reinforcement Learning

**Hyungkyu Kang** [* 1]  **Byeongchan Kim** [* 1]  **Min-hwan Oh** [1]

## Abstract

Offline goal-conditioned reinforcement learning (GCRL) provides a practical framework for obtaining goal-reaching policies from fixed datasets. However, learning a reliable goal-conditioned value function in long-horizon tasks remains challenging. In this paper, we identify erroneous generalization in goal-conditioned value functions as a fundamental bottleneck, and demonstrate that appropriate inductive bias in the value function is crucial for addressing the bottleneck. Building on these findings, we propose Latent-Aligned Value Learning (LAVL), an offline GCRL algorithm that integrates latent-representation-based value generalization with hierarchical planning in a unified framework. Extensive experiments on OGBench demonstrate that LAVL consistently outperforms existing offline GCRL methods, achieving the highest performance on **20** out of 22 datasets. Notably, LAVL exhibits strong performance in long-horizon tasks and trajectory stitching datasets, where prior methods suffer significant performance degradation. Our code is available at https://github.com/oh-lab/LAVL.git.

## 1. Introduction

Offline goal-conditioned reinforcement learning (GCRL) aims to learn goal-reaching policies by leveraging pre-collected datasets without additional environment interaction (Levine et al., 2020; Park et al., 2025a). However, learning a reliable goal-conditioned value in long-horizon environments remains challenging (Ghosh et al., 2021; Chane-Sane et al., 2021). In particular, under sparse rewards, temporal-difference (TD) learning struggles to propagate rewards, often resulting in noisy and unreliable value estimates that induce brittle control policies (Park et al., 2023).

[1]Seoul National University, Seoul, Republic of Korea. Correspondence to: Min-hwan Oh <minoh@snu.ac.kr>.

*Proceedings of the 43$^{rd}$ International Conference on Machine Learning*, Seoul, South Korea. PMLR 306, 2026. Copyright 2026 by the author(s).

Recent empirical studies (Ke et al., 2025; Ahn et al., 2025; Giammarino et al., 2025; Giammarino & Qureshi, 2025) suggest that the core difficulty in offline GCRL cannot be attributed solely to stochastic noise or local estimation error. Instead, a fundamental bottleneck arises from the inconsistency between the learned goal-conditioned value function and the true temporal distance of the environment. Although several methods have been proposed to address this issue (Ke et al., 2025; Ahn et al., 2025; Giammarino et al., 2025; Giammarino & Qureshi, 2025), they do not fully resolve the problem: (i) their performance degrades sharply with increasing task horizon, (ii) trajectory stitching remains challenging, and (iii) some approaches introduce additional value networks with extra computational and tuning costs. Crucially, there has been limited analysis about why goal-conditioned value learning fails and which components of the learning pipeline are primarily responsible.

In this work, we identify erroneous generalization in the goal-conditioned value function as a fundamental bottleneck. When value functions generalize by assigning similar values to states that are close in Euclidean distance, they often propagate value estimates across states that are distant in temporal distance, leading to large estimation errors. We demonstrate that the inductive bias in the value function architecture plays a central role in guiding accurate value generalization. While quasimetric architectures (Liu et al., 2023; Wang & Isola, 2022) introduce useful inductive bias, their performance varies substantially and can even degrade in certain tasks. These findings motivate a new offline GCRL method that achieves reliable value generalization and consistent performance across diverse environments.

In response, we propose **Latent-Aligned Value Learning (LAVL)**, an offline GCRL algorithm built on a new value function architecture, **Latent Alignment Network (LAN)**. LAN parameterizes the goal-conditioned value as the distance between learned states and goal representations, allowing value estimates to generalize through latent alignment rather than Euclidean proximity in the state space. Across a wide range of tasks, LAN-parameterized value functions improve upon or match existing architectures. This implies that the latent representation in LAN is the key to harnessing value generalization, rather than quasimetric structures.

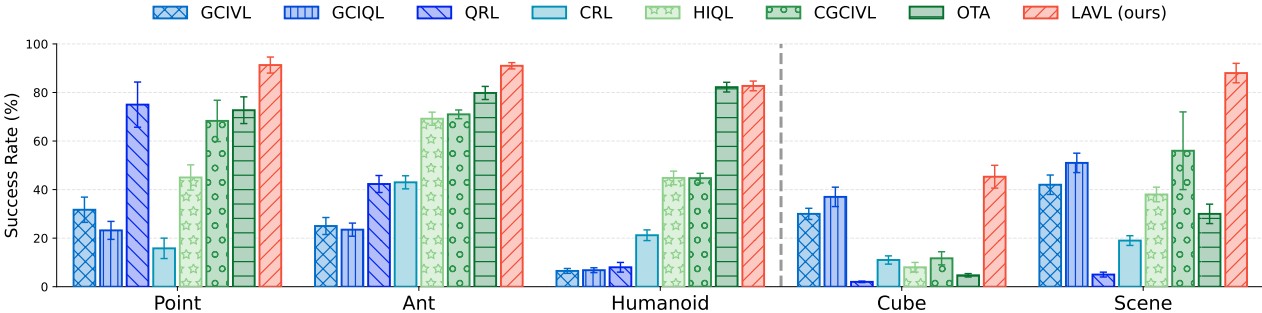

*Figure 1.* Success rates on OGBench maze navigation (`Point`, `Ant`, and `Humanoid`) and robot manipulation (`Cube` and `Scene`) tasks. For the maze environments, we report the average success rate across three maze sizes (`medium`/`large`/`giant`) and two dataset types (`navigate`/`stitch`). For `Cube`, results are averaged over `play` datasets (`single`/`double`/`triple`), and `Scene` reports the success rate on the `play` dataset. Each task setting is evaluated with 8 random seeds.

Building on LAN, our proposed algorithm, LAVL, is developed to fully exploit its strengths in long-horizon GCRL. To preserve global Bellman consistency while stabilizing the local value landscape, LAVL augments the TD objective with a continuity regularization term that suppresses sharp local fluctuations in the learned value function. In addition, LAVL naturally incorporates LAN into a hierarchical policy framework, facilitating effective long-horizon control. Together, these components enable LAVL to combine effective generalization, stable value learning, and hierarchical planning under a unified framework.

As Figure 1 demonstrates, LAVL consistently outperforms existing offline GCRL methods on OGBench (Park et al., 2025a), from maze navigation to robotic manipulation. Notably, LAVL maintains strong performance in long-horizon tasks and datasets that require extensive trajectory stitching, where existing methods suffer significant performance degradation. Moreover, LAVL substantially outperforms prior hierarchical methods in robotic manipulation tasks, demonstrating that LAN enables robust subgoal representation learning for hierarchical policies.

Our contributions are summarized as follows:

- We identify erroneous generalization in value functions as a fundamental bottleneck of goal-conditioned value learning, where values propagate by Euclidean proximity instead of temporal distance. To the best of our knowledge, our work is the first to demonstrate that inductive bias in the value function is crucial for reliable value generalization.

- We propose LAVL, an offline GCRL algorithm that integrates effective generalization, stable value learning, and hierarchical planning within a unified framework. Through latent alignment between states and goal representations, LAVL enables reliable value generalization and stable long-horizon learning.

- Extensive numerical experiments on OGBench demonstrate that LAVL consistently outperforms existing offline GCRL methods, achieving the best performance on **20** out of 22 datasets. Notably, LAVL maintains strong performance on long-horizon tasks and datasets that require extensive trajectory stitching.

## 2. Related Works

**GCRL.** GCRL aims to learn a goal-conditioned policy that can achieve any goal in the goal space (Schaul et al., 2015). We focus on the offline GCRL setting, where the agent learns from pre-collected trajectories without online interaction. Goal-conditioned behavior cloning approaches directly imitate actions in the dataset (Ghosh et al., 2021; Yang et al., 2022). This approach is simple, but it cannot stitch trajectories, and the performance largely depends on the quality of the offline dataset. Like standard RL, many approaches are based on learning a goal-conditioned value function. The value-based approaches include TD learning (Kostrikov et al., 2022; Park et al., 2023; Giammarino et al., 2025; Giammarino & Qureshi, 2025; Ke et al., 2025; Ahn et al., 2025; Myers et al., 2025), state occupancy matching (Durugkar et al., 2021; Ma et al., 2022), contrastive learning (Eysenbach et al., 2021; 2022; Myers et al., 2024; 2025), and constrained optimization (Wang et al., 2023).

GCRL raises major challenges due to the sparse reward signal and long horizon. To address these issues, hindsight relabeling (Andrychowicz et al., 2017; Chebotar et al., 2021) methods utilize in-trajectory future states as goals, encouraging reward propagation. In addition, hierarchical RL approaches have been studied for planning with a long horizon, by subgoal generation (Kulkarni et al., 2016; Vezhnevets et al., 2017; Nachum et al., 2018; Nasiriany et al., 2019; Nair & Finn, 2020; Gürtler et al., 2021; Chane-Sane et al., 2021; Park et al., 2023; Ahn et al., 2025) or graph-based planning (Eysenbach et al., 2019; Huang et al., 2019;

Kim et al., 2021; Zhang et al., 2021).

**Inductive Bias for GCRL.** Since the optimal goal-conditioned value function is quasimetric (Wang et al., 2023), some of the prior GCRL works employ specialized quasimetric architectures, such as Metric Residual Network (MRN) (Liu et al., 2023) or Interval Quasimetric Embedding (IQE) (Wang & Isola, 2022), to parameterize the value function (Liu et al., 2023; Wang et al., 2023; Myers et al., 2024; 2025). The quasimetric architectures have been applied to TD learning (Liu et al., 2023; Myers et al., 2025), constrained optimization (Wang et al., 2023), and contrastive learning (Myers et al., 2024; 2025), validating their empirical advantage for value learning. Further discussion on quasimetric architectures is presented in Appendix C.1.

The importance of inductive bias is also pronounced in representation learning for GCRL. Prior works proposed metric-based representation learning (Sermanet et al., 2018; Ma et al., 2023; Park et al., 2024), where a metric embedding for states is trained by TD learning (Park et al., 2024), dual RL (Ma et al., 2023), or contrastive learning (Sermanet et al., 2018). We note that they utilized the metric structure to obtain the state representation used for downstream tasks, while we focus on the effectiveness of architectural inductive bias for reliable value generalization.

## 3. Preliminaries

We consider offline GCRL defined by a discounted Markov decision process (MDP) $(\mathcal{S}, \mathcal{A}, \mathcal{G}, p_0, p, r, \gamma)$ and a dataset $\mathcal{D}$, where $\mathcal{S}$ denotes the state space, $\mathcal{A}$ denotes the action space, $\mathcal{G}$ denotes the goal space, $p_0$ denotes the initial state distribution, $p(\cdot|s, a)$ is the transition dynamics for $(s, a) \in \mathcal{S} \times \mathcal{A}$, $r(s, g)$ is the goal-conditioned reward for $(s, g) \in \mathcal{S} \times \mathcal{G}$, and $\gamma \in (0, 1)$ is the discount factor. Although the goal space $\mathcal{G}$ can be defined independently of the state space $\mathcal{S}$, we focus mainly on the state-reaching problem where $\mathcal{G} = \mathcal{S}$. The offline dataset $\mathcal{D}$ consists of trajectories $\tau = (s_0, a_0, s_1, a_1, \ldots, s_T)$ collected by some behavior policy $\mu$. The goal of the agent is to maximize the expected cumulative reward $J(\pi) := \mathbb{E}_{g \sim q, \tau \sim p^\pi} [\sum_{t=0}^{T} \gamma^t r(s_t, g)]$ where $q$ is a goal distribution and $p^\pi(\tau) := p_0(s_0) \Pi_{t=0}^{T-1} \pi(a_t|s_t) p(s_{t+1}|s_t, a_t)$ is the trajectory distribution induced by $\pi$.

## 4. Motivation: Bottleneck of Goal-Conditioned Value Learning

In this section, we empirically analyze the bottlenecks of existing offline GCRL algorithms, focusing on the interplay between value function architectures and learning objectives. We begin by identifying *overgeneralization*, a failure mode in which learned goal-conditioned values generalize value

estimates in undesirable ways. Through ablation studies, we observe that incorporating architectural inductive bias into the value function mitigates overgeneralization. Additionally, quasimetric methods, despite their inductive bias in architectures, exhibit substantial performance variability across environments and frequently result in catastrophic training failures. Taken together, these findings motivate the development of an offline GCRL method that achieves both reliable value generalization and consistent performance across diverse tasks.

### 4.1. Overgeneralization

As illustrated in the left panel of Figure 2, we visualize the goal-conditioned value functions learned by GCIVL (Park et al., 2023) and QRL (Wang et al., 2023) on a maze navigation task. Note that GCIVL uses the multi-layer perceptron (MLP) value network, whereas QRL uses the IQE (Wang & Isola, 2022) architecture, which is a quasimetric model. We observe that GCIVL with MLP assigns incorrectly high value estimates to states that are close to the goal in Euclidean distance, as if the value is leaked through the maze walls. This phenomenon is basically an erroneous generalization of value estimates, so we call it *overgeneralization*. However, QRL with IQE produces remarkably consistent value landscapes without overgeneralization, where the value propagates coherently with temporal distance. Hence, the following research question arises:

**Question 1**: *What drives overgeneralization in the goal-conditioned value function? Is it the value function architecture or the learning objective?*

We further conduct ablation experiments in the middle panel of Figure 2, instantiating GCIVL with IQE (instead of MLP) and QRL with MLP (instead of IQE), to disentangle the contribution of the value function architectures and the learning objectives. QRL with MLP fails to estimate goal-conditioned values, leading to severe instability. In contrast, GCIVL with IQE mitigates the overgeneralization observed in GCIVL with MLP. Therefore, the advantage of the original QRL comes from its architecture, not from the learning objective. We note that the robustness against overgeneralization is not the exclusive property of IQE. Value function architectures with appropriate inductive bias enjoy the advantage (See Section B.5 for details).

> **Finding 1: Structural Inductive Bias**
>
> While the standard MLP-parameterized value function suffers from value overgeneralization, value functions with **structural inductive bias** can mitigate overgeneralization.

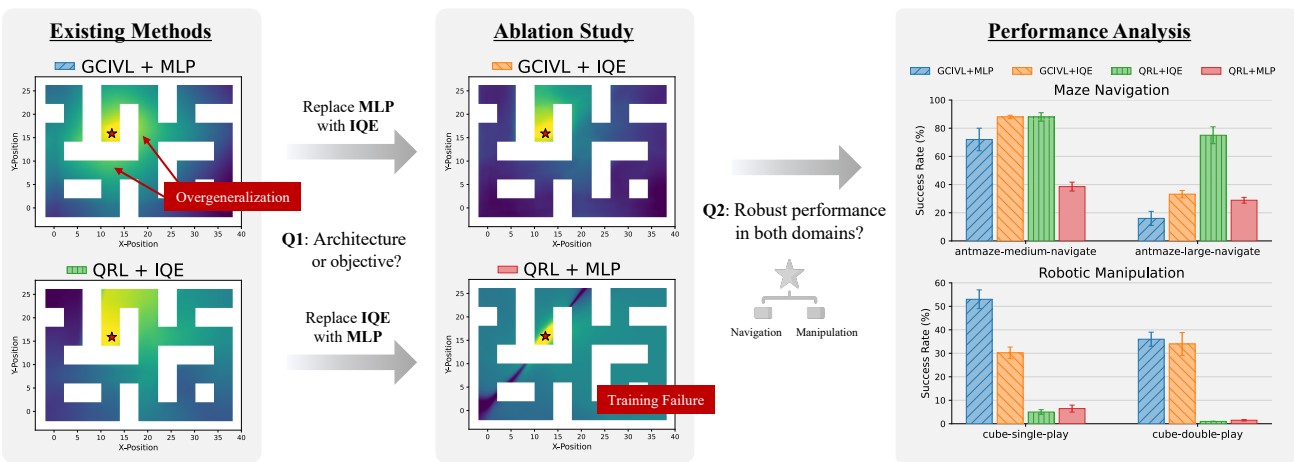

*Figure 2.* (Left) Visualization of learned goal-conditioned values on `antmaze-large-stitch`. The red star denotes the goal position. The GCIVL agent with MLP-parameterized value exhibits incorrect generalization, while QRL with IQE does not. (Middle) Ablation experiments on value function architecture. The inductive bias of IQE effectively mitigates overgeneralization. (Right) Performance in maze navigation and robotic manipulation tasks. No single method shows consistently high performance.

## 4.2. Limitations of Quasimetric Methods

By visualizing the value landscape in the maze navigation task, we observed that the mitigation of overgeneralization primarily stems from the inductive bias of the IQE architecture, rather than from the learning objective of QRL. While this visualization provides valuable intuition, it does not necessarily imply that such advantages translate into consistent performance gains. Moreover, since our analysis thus far has focused exclusively on maze navigation, it remains unclear whether the quasimetric methods generalize to other domains, particularly robotic manipulation. These observations naturally lead to the following research question:

**Question 2**: *Do IQE-based methods achieve high performance in both maze navigation and robotic manipulation?*

In the right panel of Figure 2, the results in maze navigation and robotic manipulation answer our question *negatively*.

**Limitation of QRL.** QRL with IQE achieves strong performance in maze navigation tasks, which is consistent with the well-structured value landscape observed earlier. However, its performance collapses in robotic manipulation tasks, yielding near-zero success rates, even in settings where GCIVL with MLP learns reliably. A similar failure mode is observed for QRL with MLP, whereas GCIVL with IQE does not exhibit such catastrophic degradation. These results indicate that the large performance discrepancy between maze navigation and robotic manipulation is primarily attributable to the QRL objective itself, rather than to the choice of value function architecture.

**Limitation of IQE.** Replacing the MLP with IQE in GCIVL consistently improves performance in maze navigation tasks compared to GCIVL with MLP, confirming that overgeneral-

ization indeed constitutes a key bottleneck in value learning. Importantly, unlike QRL, this combination does not lead to complete training failure in robotic manipulation tasks. From this perspective, GCIVL with IQE can be viewed as a reasonable compromise between standard GCIVL and QRL.

Nevertheless, GCIVL with IQE is still far from a satisfactory solution. Its performance gains in maze navigation are insufficient to match those of QRL with IQE, while in robotic manipulation tasks, the IQE architecture tends to degrade performance relative to simpler MLP-based parameterizations. These results suggest that, although IQE alleviates overgeneralization, its inductive bias may be misaligned with the requirements of manipulation tasks.

---

**Finding 2: Limitations of Quasimetric Method**

Our experiments lead to two key conclusions.
(i) The QRL objective induces severe training instability in robotic manipulation tasks, highlighting **the necessity of TD learning** for achieving robust performance across diverse domains.
(ii) While GCIVL with IQE improves upon GCIVL with MLP by mitigating overgeneralization, **the IQE itself can degrade performance** in some tasks.

---

Finally, these findings lead to our motivation:

*Can we develop an offline GCRL algorithm that addresses value overgeneralization and achieves consistently high performance across diverse tasks?*

In the following section, we build on this motivation by combining a new value function architecture with a hierarchical policy framework.

# 5. Latent-Aligned Value Learning

In this section, we propose **Latent-Aligned Value Learning (LAVL)**, an offline GCRL algorithm leveraging a new value function architecture and hierarchical policy framework. We first introduce our proposed network architecture.

## 5.1. Latent Alignment Network

We propose **Latent Alignment Network (LAN)**, a new network architecture that aligns goal-conditioned value estimates with the geometry of the underlying latent space. The core design of LAN is to represent the goal-conditioned value as the negative Euclidean distance in a learned latent space. Specifically, we parameterize the value function as:

$$V(s,g) = -\|\varphi_S(s) - \varphi_G(g)\|_2,$$

where $\varphi_S : \mathcal{S} \to \mathbb{R}^d$ and $\varphi_G : \mathcal{G} \to \mathbb{R}^d$ are the state and goal representation networks, respectively. This structure is conceptually motivated by the metric-based state representation (Sermanet et al., 2018; Ma et al., 2023) and the Hilbert representation (Park et al., 2024), where a single state embedding $\varphi$ is trained by approximate temporal distance by $\|\varphi(s) - \varphi(g)\|_2$. The key distinction of LAN lies in its asymmetric representations ($\varphi_S \neq \varphi_G$). By breaking the strict metric structure, this asymmetry introduces additional flexibility that proves crucial for achieving strong performance with both flat policies (see Figure 3) and hierarchical policies (see Figure 6).

**Remark.** The LAN architecture differs fundamentally from prior metric-based representations (Sermanet et al., 2018; Ma et al., 2023; Park et al., 2024) in both design objective and empirical behavior. First, metric and Hilbert representations are primarily introduced to learn reusable state embeddings for downstream tasks such as planning. In contrast, LAN is explicitly designed as a *value function architecture*, where latent representations are employed to impose inductive bias on value generalization. Second, LAN relaxes the metric constraints that may be overly restrictive for value learning, while retaining sufficient structure to guide accurate generalization. The empirical advantages of this design choice are validated in Figure 3 and Figure 6, where LAN consistently outperforms existing architectures.

The advantage of LAN is evident in Figure 3, where GCIVL with LAN outperforms alternative architectures, including quasimetric architectures (IQE and MRN), the metric-based parameterization (Hilbert), and MLP. Moreover, these performance gains are consistent across diverse tasks, addressing the limitations of existing value function architectures. Furthermore, LAN enjoys improved optimization stability, which we analyze in detail in Section B.3.

The superior performance of LAN over quasimetric architectures implies that *the asymmetric latent representation is*

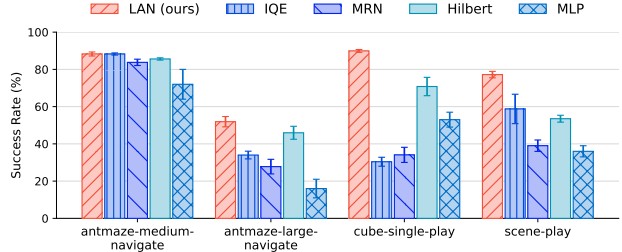

*Figure 3.* Comparison of success rates for GCIVL agents with different value function architectures across four datasets.

*key to harnessing generalization* in goal-conditioned value learning, rather than the quasimetric property. This observation is consistent with prior work showing that enforcing quasimetric constraints during training can be overly restrictive (Ke et al., 2025). Although the optimal goal-conditioned value function is quasimetric, neither intermediate value estimates nor the solution of the TD objective are required to satisfy this property (Wang et al., 2023). As a result, imposing strong constraints such as a quasimetric structure in the architecture can hinder optimization. In contrast, LAN controls generalization through learned latent representations without explicitly enforcing a quasimetric structure, achieving superior empirical performance.

## 5.2. Value Learning with LAN

We train the LAN-parameterized value function $V(s,g)$ by minimizing the following IVL objective (Park et al., 2023):

$$\mathcal{L}_{\text{TD}}(V) = \mathbb{E}_{\substack{(s,s')\in\mathcal{D}, \\ g\sim p^{\mathcal{D}}_{\text{mixed}}(\cdot|s)}} [\ell^{\kappa}_2(r(s,g)+\gamma\tilde{V}(s',g)-V(s,g))],$$

where $\ell^{\kappa}_2(x) = |\kappa - \mathbb{1}\{x < 0\}|x^2$ denotes the expectile loss, $\tilde{V}$ is target value, and $p^{\mathcal{D}}_{\text{mixed}}(\cdot|s)$ is the goal sampling distribution (See Appendix A.3). While the LAN value trained solely with the IVL objective achieves global Bellman consistency, value learning in long-horizon settings can still suffer from localized instability. In particular, sparse reward signals must propagate through a large number of steps up to the task horizon. During this process, the Bellman objective alone leaves the local geometry of the value function under-constrained, which may lead to sharp local fluctuations in the value function. To stabilize value learning, we introduce a local continuity regularization term:

$$\mathcal{L}_{\text{Reg}}(V) = \mathbb{E}_{\substack{(s,s')\in\mathcal{D}, \\ g\sim p^{\mathcal{D}}_{\text{rand}}(\cdot|s)}} \left[ \left( (V(s,g)-V(s',g))^2 - \delta^2 \right)_+ \right],$$

where $p^{\mathcal{D}}_{\text{rand}}$ is the random goal distribution, and $(x)_+ := \mathbb{1}\{x > 0\}x$. This term penalizes value variation between neighboring states when it exceeds a threshold $\delta$, thereby complementing the IVL objective by biasing learning toward locally stable value functions while preserving global Bellman consistency.

This idea is related to recent work (Giammarino et al., 2025), which regularizes the value function by encouraging its gradient norm $\|\nabla_s V(s, g)\|$ to remain constant, thereby stabilizing the local geometry. In contrast, our approach enforces local regularity via finite differences on sampled transitions, avoiding costly gradient computation. Combining the TD objective with our regularization term, we obtain the following total loss:

$$\mathcal{L}(V) = \mathcal{L}_{\text{TD}}(V) + w_c \mathcal{L}_{\text{Reg}}(V)$$

where $w_c \geq 0$ is a continuity regularization hyperparameter. Although $\delta$ can be treated as a hyperparameter, we find that setting $\delta = 1 + (1 - \gamma)|\bar{V}|$ works well across tasks, where $\bar{V}$ is the batch mean of the value estimates.

### 5.3. Hierarchical Policy Extraction in LAVL

We adopt the hierarchical policy extraction framework of HIQL (Park et al., 2023), which consists of a high-level policy $\pi^h(w|s, g)$ and a low-level policy $\pi^l(a|s, w)$. The high-level policy generates a subgoal representation $w$ conditioned on the current state $s$ and the final goal $g$, and the low-level policy outputs actions conditioned on the current state $s$ and subgoal representation $w$.

While LAVL follows the overall hierarchical structure of HIQL, it differs in the parameterization of the subgoal representation. In the original formulation of HIQL, the subgoal representation is defined as a goal-only mapping, whereas the implementation in HIQL adopts a state-dependent representation $w = \phi([g, s])$ to improve empirical performance. In LAVL, we adopt a goal-only subgoal representation $w = \phi(g)$, consistent with the original formulation of HIQL. This design choice maintains closer alignment with the theoretical framework underlying HIQL, and integrates naturally with the LAN parameterization of the value function as

$$V(s, g) = -\|\varphi_S(s) - \varphi_G(\phi(g))\|_2.$$

From this perspective, the subgoal representation in LAVL can be viewed as an information bottleneck applied before the final goal representation in LAN. In Section 6, we demonstrate that the combination of LAN and the goal-only subgoal representation leads to substantial performance improvements in practice.

Both policies are trained using Advantage-Weighted Regression (AWR) (Peng et al., 2019):

$$\mathcal{L}(\pi^h) = \mathbb{E}_{\substack{(s_t, s_{t+k}) \in \mathcal{D}, \\ g \sim p_{\text{mixed}}^{\mathcal{D}}(\cdot|s_t)}} \left[-e^{\beta^h A^h(s_t, s_{t+k}, g)} \right.$$
$$\left. \log \pi^h(\phi(s_{t+k}) \mid s_t, g)\right],$$
$$\mathcal{L}(\pi^l) = \mathbb{E}_{(s_t, a_t, s_{t+1}, s_{t+k}) \in \mathcal{D}}[-e^{\beta^l A^l(s_t, s_{t+1}, s_{t+k})}$$
$$\log \pi^l(a_t \mid s_t, \phi(s_{t+k}))],$$

where $A^h(s_t, s_{t+k}, g) = V(s_{t+k}, g) - V(s_t, g)$ and $A^l(s_t, s_{t+1}, s_{t+k}) = V(s_{t+1}, s_{t+k}) - V(s_t, s_{t+k})$ denote the high-level and low-level advantages, respectively, and $\beta^h$ and $\beta^l$ are the corresponding temperature parameters.

## 6. Experiments

### 6.1. Experimental Setup

**Datasets.** We evaluate our proposed algorithm and baselines on OGBench (Park et al., 2025a), a comprehensive benchmark for offline GCRL. We use **22 datasets** encompassing long-horizon maze navigation tasks (Pointmaze, Antmaze, and Humanoidmaze) and robotic manipulation tasks (Cube and Scene). The maze tasks evaluate the agent's long-horizon reasoning ability with extremely long episode lengths, and the robotic manipulation tasks require precise object manipulation with a 6-DoF robot arm. More details on the datasets are included in Appendix A.1.

**Baselines.** We compare LAVL to prior offline GCRL algorithms: (i) GCBC (Ghosh et al., 2021) is a goal-conditioned behavioral cloning using in-trajectory future states as goals. (ii) GCIQL and (iii) GCIVL (Kostrikov et al., 2022; Park et al., 2023) estimate optimal {Q,V}-function via expectile regression on Bellman targets. (iv) QRL (Wang et al., 2023) performs constrained optimization on a quasimetric value network such as IQE (Wang & Isola, 2022). (v) CRL (Eysenbach et al., 2022) is based on the policy evaluation via contrastive learning, followed by iterative policy improvement. (vi) HIQL (Park et al., 2023) employs hierarchical policy extraction, where the value function is trained by IVL. (vii) CGCIVL (Ke et al., 2025) and (viii) OTA (Ahn et al., 2025) improve HIQL by enhancing value estimation. CGCIVL employs conservatism regularization and IQE-based value distillation, while OTA reduces the effective horizon by option-based temporal abstraction. Additionally, we compare Eik-HIQL (Giammarino et al., 2025) and Eik-HiQRL (Giammarino & Qureshi, 2025) in Appendix B.2, as their evaluation protocol differs from OGBench.

### 6.2. Benchmark Results

Table 1 summarizes the experimental results across the full benchmark. LAVL achieves the best performance in **20** out of 22 tasks, consistently outperforming all baselines. Based on these results, we draw several key observations below.

**Robotic Manipulation.** For robotic manipulation tasks such as Cube and Scene, we observe markedly different performance across algorithms. Although QRL achieves high performance in maze navigation, it attains very low success rates on manipulation tasks. Similarly, while HIQL often improves over GCIVL on maze tasks, these gains do not consistently transfer to manipulation tasks and can even degrade performance on certain tasks (e.g., Cube). In

*Table 1.* **Full benchmark table.** We report the average (binary) success rate (%) across the five test-time goals on each task, averaged over 8 random seeds. † The results of CGCIVL are reproduced by the official implementation. Details are explained in Appendix B.1. ‡The results of OTA in robotic manipulation tasks (Cube and Scene) are reproduced by the official implementation.

| Environment | Dataset Type | Non-hierarchical | | | | | Hierarchical | | | |
|---|---|---|---|---|---|---|---|---|---|---|
| | | GCBC | GCIVL | GCIQL | QRL | CRL | HIQL | CGCIVL† | OTA‡ | LAVL (ours) |
| Pointmaze | medium-navigate-v0 | $9_{\pm6}$ | $63_{\pm6}$ | $53_{\pm8}$ | $82_{\pm5}$ | $29_{\pm7}$ | $79_{\pm5}$ | $76_{\pm4}$ | $86_{\pm2}$ | $\mathbf{92}_{\pm1}$ |
| | large-navigate-v0 | $29_{\pm6}$ | $45_{\pm5}$ | $34_{\pm3}$ | $86_{\pm9}$ | $39_{\pm7}$ | $58_{\pm5}$ | $77_{\pm10}$ | $85_{\pm5}$ | $\mathbf{93}_{\pm3}$ |
| | giant-navigate-v0 | $1_{\pm2}$ | $0_{\pm0}$ | $0_{\pm0}$ | $68_{\pm7}$ | $27_{\pm10}$ | $46_{\pm9}$ | $65_{\pm10}$ | $72_{\pm6}$ | $\mathbf{91}_{\pm4}$ |
| | medium-stitch-v0 | $23_{\pm18}$ | $70_{\pm14}$ | $21_{\pm9}$ | $80_{\pm12}$ | $0_{\pm1}$ | $74_{\pm6}$ | $66_{\pm5}$ | $75_{\pm5}$ | $\mathbf{90}_{\pm2}$ |
| | large-stitch-v0 | $7_{\pm5}$ | $12_{\pm6}$ | $31_{\pm2}$ | $84_{\pm15}$ | $0_{\pm0}$ | $13_{\pm6}$ | $79_{\pm3}$ | $66_{\pm8}$ | $\mathbf{87}_{\pm8}$ |
| | giant-stitch-v0 | $0_{\pm0}$ | $0_{\pm0}$ | $0_{\pm0}$ | $50_{\pm8}$ | $0_{\pm0}$ | $0_{\pm0}$ | $47_{\pm19}$ | $52_{\pm7}$ | $\mathbf{95}_{\pm2}$ |
| Antmaze | medium-navigate-v0 | $29_{\pm4}$ | $72_{\pm8}$ | $71_{\pm4}$ | $88_{\pm3}$ | $95_{\pm1}$ | $96_{\pm1}$ | $94_{\pm1}$ | $96_{\pm1}$ | $\mathbf{97}_{\pm1}$ |
| | large-navigate-v0 | $24_{\pm2}$ | $16_{\pm5}$ | $34_{\pm4}$ | $75_{\pm6}$ | $83_{\pm4}$ | $91_{\pm2}$ | $90_{\pm1}$ | $92_{\pm1}$ | $\mathbf{93}_{\pm1}$ |
| | giant-navigate-v0 | $0_{\pm0}$ | $0_{\pm0}$ | $0_{\pm0}$ | $14_{\pm3}$ | $16_{\pm3}$ | $65_{\pm5}$ | $65_{\pm2}$ | $77_{\pm4}$ | $\mathbf{85}_{\pm2}$ |
| | medium-stitch-v0 | $45_{\pm11}$ | $44_{\pm6}$ | $29_{\pm6}$ | $59_{\pm7}$ | $53_{\pm6}$ | $94_{\pm1}$ | $88_{\pm2}$ | $93_{\pm1}$ | $\mathbf{97}_{\pm1}$ |
| | large-stitch-v0 | $3_{\pm3}$ | $18_{\pm2}$ | $7_{\pm2}$ | $18_{\pm2}$ | $11_{\pm2}$ | $67_{\pm5}$ | $81_{\pm2}$ | $84_{\pm3}$ | $\mathbf{92}_{\pm1}$ |
| | giant-stitch-v0 | $0_{\pm0}$ | $0_{\pm0}$ | $0_{\pm0}$ | $0_{\pm0}$ | $0_{\pm0}$ | $2_{\pm2}$ | $8_{\pm3}$ | $37_{\pm6}$ | $\mathbf{82}_{\pm2}$ |
| Humanoidmaze | medium-navigate-v0 | $8_{\pm2}$ | $24_{\pm2}$ | $27_{\pm2}$ | $21_{\pm8}$ | $60_{\pm4}$ | $89_{\pm2}$ | $88_{\pm1}$ | $\mathbf{94}_{\pm1}$ | $\mathbf{94}_{\pm1}$ |
| | large-navigate-v0 | $1_{\pm0}$ | $2_{\pm1}$ | $2_{\pm1}$ | $5_{\pm1}$ | $24_{\pm4}$ | $49_{\pm4}$ | $46_{\pm2}$ | $\mathbf{83}_{\pm2}$ | $74_{\pm3}$ |
| | giant-navigate-v0 | $0_{\pm0}$ | $0_{\pm0}$ | $0_{\pm0}$ | $1_{\pm0}$ | $3_{\pm2}$ | $12_{\pm4}$ | $10_{\pm2}$ | $\mathbf{92}_{\pm1}$ | $83_{\pm2}$ |
| | medium-stitch-v0 | $29_{\pm5}$ | $12_{\pm2}$ | $12_{\pm3}$ | $18_{\pm2}$ | $36_{\pm2}$ | $88_{\pm2}$ | $89_{\pm1}$ | $88_{\pm2}$ | $\mathbf{93}_{\pm2}$ |
| | large-stitch-v0 | $6_{\pm3}$ | $1_{\pm1}$ | $0_{\pm0}$ | $3_{\pm1}$ | $4_{\pm1}$ | $28_{\pm3}$ | $20_{\pm2}$ | $57_{\pm3}$ | $\mathbf{72}_{\pm3}$ |
| | giant-stitch-v0 | $0_{\pm0}$ | $0_{\pm0}$ | $0_{\pm0}$ | $0_{\pm0}$ | $0_{\pm0}$ | $3_{\pm2}$ | $15_{\pm4}$ | $79_{\pm3}$ | $\mathbf{80}_{\pm1}$ |
| Cube | single-play-v0 | $6_{\pm2}$ | $53_{\pm4}$ | $68_{\pm6}$ | $5_{\pm1}$ | $19_{\pm2}$ | $15_{\pm3}$ | $23_{\pm4}$ | $9_{\pm1}$ | $\mathbf{83}_{\pm4}$ |
| | double-play-v0 | $1_{\pm1}$ | $36_{\pm3}$ | $40_{\pm5}$ | $1_{\pm0}$ | $10_{\pm2}$ | $6_{\pm2}$ | $7_{\pm2}$ | $3_{\pm0}$ | $\mathbf{42}_{\pm8}$ |
| | triple-play-v0 | $1_{\pm1}$ | $1_{\pm0}$ | $3_{\pm1}$ | $0_{\pm0}$ | $4_{\pm1}$ | $3_{\pm1}$ | $5_{\pm2}$ | $2_{\pm1}$ | $\mathbf{11}_{\pm2}$ |
| Scene | play-v0 | $5_{\pm1}$ | $42_{\pm4}$ | $51_{\pm4}$ | $5_{\pm1}$ | $19_{\pm2}$ | $38_{\pm3}$ | $56_{\pm16}$ | $30_{\pm4}$ | $\mathbf{88}_{\pm4}$ |

*Table 2.* Average success rate (%) for different maze sizes in OGBench. Relative Drop (Rel. Drop) denotes the percentage decrease in success rate from medium to giant.

| Algorithm | Medium | Large | Giant | Rel. Drop (%) ↓ |
|---|---|---|---|---|
| **LAVL (ours)** | **94** | **85** | **85** | **9.6**% |
| OTA | 89 | 78 | 68 | 23.1% |
| CGCIVL | 84 | 66 | 35 | 58.3% |
| QRL | 58 | 45 | 22 | 61.7% |
| HIQL | 87 | 51 | 21 | 75.4% |
| GCIVL | 48 | 16 | 0 | 100.0% |

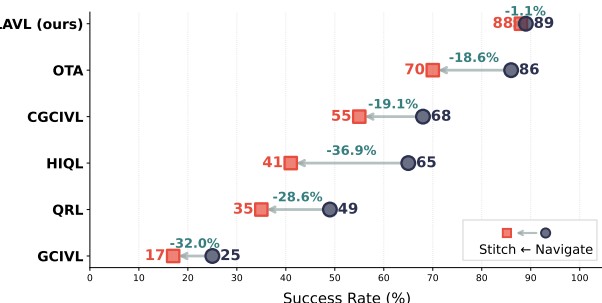

*Figure 4.* Average success rates on maze navigate and stitch datasets. The relative performance drop in stitch datasets compared to navigate datasets is indicated.

contrast, LAVL consistently outperforms HIQL and OTA by a substantial margin across robotic manipulation tasks. Moreover, LAVL also surpasses GCIVL, suggesting that its hierarchical policy helps planning in manipulation settings rather than impeding it.

**Long-Horizon Tasks.** As summarized in Table 2, while baselines suffer severe performance degradation as the maze scale increases from medium to giant, LAVL exhibits remarkable stability. Notably, LAVL maintains an 85% success rate in giant mazes with a marginal relative drop of only **9.6**%, whereas other baselines such as OTA, CG-CIVL, and HIQL experience substantial collapses of 23.1%, 58.3%, and 75.4%, respectively. These results demonstrate that LAVL maintains a robust success rate as the horizon increases, with only minimal performance degradation.

**Stitching Trajectories.** The stitch datasets consist of short trajectories, thus they require composing multiple segments to propagate reward signals over the full horizon. As illustrated in Figure 4, all baseline methods exhibit substantial degradation when trajectory stitching is required. For instance, OTA, CGCIVL, and HIQL exhibit average relative drops of 18.6%, 19.1%, and 36.9%, respectively. In contrast, LAVL shows a much smaller drop (**1.1**%), indicating improved robustness to trajectory fragmentation and stitching. These results highlight that LAVL is less sensitive to dataset-induced horizon truncation, which is critical for long-horizon goal reaching under offline data constraints.

## 6.3. Effect of Hyperparameters

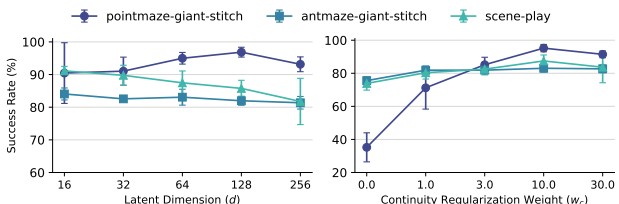

*Figure 5.* Effect of hyperparameters: (Left) Latent dimension of LAN and (Right) Continuity regularization weight.

**Latent Dimension.** Since the effectiveness of LAN relies on latent-space generalization, one natural concern is its sensitivity to the choice of the latent dimension. As shown in the left panel of Figure 5, LAVL exhibits stable performance across a wide range of latent dimensions, from 16 to 256. This robustness simplifies practical deployment, as careful tuning of the latent dimension is unnecessary. Throughout all our experiments, we fix the latent dimension to 64 without hyperparameter tuning, and LAVL consistently achieves state-of-the-art performance on OGBench.

**Continuity Regularization.** As shown in the right panel of Figure 5, continuity regularization yields clear performance gains on challenging tasks, by stabilizing local fluctuation in the value function. Its effect is most pronounced on `pointmaze-giant`, where the success rate increases from 35% to 95%, and it also improves performance on `antmaze-giant` and `scene`. Overall, the results demonstrate that continuity regularization improves performance in a task-dependent manner.

## 6.4. Ablation of Value Function Architecture

While Section 5.1 demonstrated the empirical advantage of LAN for value learning in a non-hierarchical framework, it is important to examine how LAN compares with alternative architectures in a hierarchical framework. To this end, we construct LAVL variants by replacing LAN with IQE, MRN, or Hilbert (metric parameterization) while keeping all other components unchanged (see Section C for details).

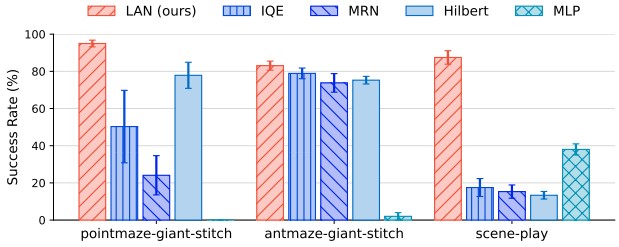

*Figure 6.* Comparison of success rates for LAVL agents with different value function architectures across three datasets. The MLP variant denotes the HIQL baseline.

Figure 6 compares LAVL with its architectural variants. The MLP variant denotes the HIQL baseline. On maze navigation tasks, the IQE, MRN, and Hilbert variants outperform the MLP baseline, and achieve performance comparable to LAN in `antmaze-giant`. These results suggest that architectural inductive biases can improve long-horizon value learning, supporting our findings in Section 4.1. However, in the `scene` dataset, LAN achieves an 88% success rate, whereas IQE, MRN, and Hilbert remain below 20%, and the MLP baseline stays below 40%. Overall, these results indicate that LAN outperforms existing quasimetric architectures within the same hierarchical framework.

## 6.5. Hierarchical Policy Framework

We follow the original HIQL design, in which a single value function is used to derive advantage estimates for both the high-level and low-level policies (Park et al., 2023). In contrast, OTA (Ahn et al., 2025) and Eik-HiQRL (Giammarino & Qureshi, 2025) train a separate high-level value function to estimate high-level advantages, while maintaining HIQL's low-level policy and subgoal representation.

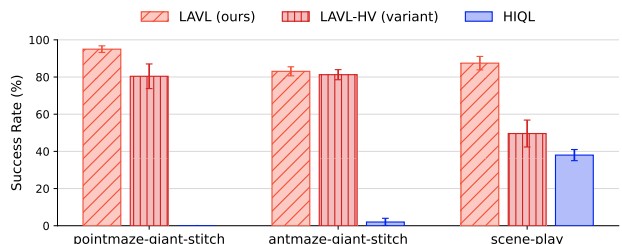

*Figure 7.* Comparison of a unitary value function and separate high-/low-level value functions for hierarchical policy extraction.

To explicitly compare these two design paradigms, we implement a variant of LAVL, termed **LAVL-HV** (Hierarchical Value), which uses separate value functions for the high-level and low-level policies. In Figure 7, LAVL-HV substantially outperforms HIQL, improving success rates from nearly 0% to over 80% for the two maze datasets. Given that LAVL-HV differs from HIQL only by replacing the high-level value function, this result indicates that the value function of LAVL provides a more informative high-level learning signal for high-level policy learning. Moreover, LAVL consistently outperforms LAVL-HV, suggesting that a unitary LAVL value function also benefits low-level policy learning through tighter coupling with the subgoal representation. As a result, LAVL eliminates the need for an additional low-level value function, avoiding the associated computational overhead and hyperparameter tuning, while achieving superior performance.

# 7. Conclusion

In this work, we show that erroneous generalization that fails to reflect temporal distance is a central bottleneck in offline GCRL, and that value generalization can be substantially improved by incorporating appropriate inductive bias into the value function. Motivated by this observation, we design the Latent Alignment Network (LAN), an effective architecture for goal-conditioned value learning, and propose Latent-Aligned Value Learning (LAVL), an algorithm that integrates LAN-based value learning with a hierarchical policy framework. Through extensive experiments on OG-Bench, we demonstrate that LAVL improves upon existing offline GCRL algorithms by a large margin, while enabling highly stable learning, particularly in long-horizon tasks and datasets that require trajectory stitching.

Our finding highlights the broader importance of value function architecture beyond offline GCRL. In particular, value-based RL algorithms may benefit from architectural inductive biases, making this an interesting direction for future work. While this work focuses on value learning and adopts the hierarchical policy extraction framework of HIQL, policy extraction for long-horizon tasks with high-dimensional action spaces remains challenging and may provide orthogonal gains. Finally, although we evaluated algorithms on OG-Bench, extending offline GCRL algorithms such as LAVL to more complex simulations and real-world robotic control remains an open problem.

# Impact Statement

This paper aims to advance the field of offline GCRL. While offline GCRL is a promising framework for many practical decision-making problems, our work primarily focuses on simulation-based experiments, which do not raise ethical concerns.

# Acknowledgements

This work was supported by the National Research Foundation of Korea (NRF) grant and the Institute of Information & communications Technology Planning & Evaluation (IITP) grant both funded by the Korea government (MSIT) (No. RS-2022-NR071853, RS-2023-00222663, RS-2025-25463302, RS-2026-25507282).

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

# A. Experimental Details

## A.1. Details on Environments and Datasets

**PointMaze.** PointMaze uses a low-dimensional 2D point-mass agent with simple continuous dynamics. While both state space and action space are low-dimensional, this environment is challenging due to sparse rewards and a long task horizon. Moreover, the combination of a low-dimensional state space and a long horizon renders the value function and policy susceptible to overfitting. As a result, many algorithms exhibit lower performance in this environment compared to high-dimensional AntMaze or HumanoidMaze.

**AntMaze.** AntMaze consists of an 8-DoF quadrupedal Ant robot with a 29-dimensional state space, including its joint configuration and velocity. The agent must jointly solve low-level locomotion and high-level navigation, making representation learning and long-horizon credit assignment difficult.

**HumanoidMaze.** HumanoidMaze features a 21-DoF full humanoid robot with a 59-dimensional state space and complex contact-rich dynamics. The state and action spaces are large, and successful navigation requires stable locomotion, balance, and long-horizon planning, making this the most challenging maze environment in terms of control complexity.

**Cube.** Cube is a robotic manipulation task using a 6-DoF robot arm to perform pick-and-place with multiple cubes, including stacking and permutation goals. The dataset is collected in a "play" style by a scripted policy that randomly picks and places cubes, resulting in unstructured but diverse manipulation trajectories. Test-time goals require composing multiple atomic pick-and-place skills into longer plans not explicitly present in the data.

**Scene.** Scene is a long-horizon manipulation environment involving multiple interactive objects such as drawers, windows, locks, and a cube, with dependencies between actions (e.g., unlocking before opening). Play-style datasets are collected by scripted policies that randomly interact with objects, producing highly diverse but non-task-specific trajectories. Evaluation tasks require sequential reasoning and correct ordering of multiple subtasks, sometimes up to eight atomic actions.

For maze navigation environments, we have three maze sizes and two data collection methods as options:

**Maze size.** All maze environments are provided in three sizes: medium, large, and giant. Medium and large follow layouts similar to prior D4RL (Fu et al., 2020) mazes, while giant mazes substantially increase path length and require long-horizon planning, reaching up to thousands of environment steps for humanoid agents.

**Data collection.** Two main data collection protocols are used across all maze tasks. Navigate datasets are collected by a noisy expert policy that repeatedly navigates to randomly sampled goals, yielding diverse but goal-directed trajectories. Stitch datasets consist of short, local goal-reaching segments with a limited horizon, intentionally requiring algorithms to stitch multiple trajectory fragments together to reach distant goals.

## A.2. Training and Evaluation Details

We follow the standard evaluation protocol of OGBench, where we evaluate the agent at 800K, 900K, and 1M gradient steps, and then report the average success rate. The agent is evaluated 50 times for each of the five evaluation state-goal pairs, for a total of 750 evaluations per dataset (3 evaluation × 5 state-goal pairs × 50 trials). Our implementation is based on the OGBench code, and each run takes 2-4 hours (including evaluation) on an RTX 3090 GPU.

## A.3. Hyperparameters

We use the goal sampling scheme of OGBench, which is a mixture of the following four conditional distributions:

- $p_{\mathrm{cur}}^{\mathcal{D}}(g|s)$ is the Dirac delta distribution at the current state $s$. Sampling from this distribution is essential for providing nontrivial rewards.

- $p_{\mathrm{traj}}^{\mathcal{D}}(g|s)$ is a uniform in-trajectory distribution. When $s = s_t$ in a trajectory $(s_0, s_1, \ldots, s_T)$, we sample an index $k$ uniformly from $[\min(t, T-1), T-1]$ and set $g = s_k$.

- $p_{\mathrm{geom}}^{\mathcal{D}}(g|s)$ is a geometric in-trajectory distribution. When $s = s_t$ in a trajectory $(s_0, s_1, \ldots, s_T)$, we sample an index $k$ from $\mathrm{Geom}(1 - \gamma)$, and set $g = s_{\min(t+k, T-1)}$.

- $p_{\mathrm{rand}}^{\mathcal{D}}(g|s)$ is the uniform distribution over the dataset $\mathcal{D}$. Since $g$ is uniformly sampled, this distribution is independent of the condition $s$.

We denote the mixture distribution as $p^{\mathcal{D}}_{\text{mixed}}$. While the default goal sampling ratio of OGBench works well in general, sampling from $p^{\mathcal{D}}_{\text{geom}}(g|s)$ with a discount factor close to 1 concentrates most of the probability on the last state of the episode when the episode is short. Therefore, we increase the ratio of random goals and use uniform in-trajectory sampling for the maze `stitch` dataset, which consists of short trajectories.

The common hyperparameters are presented in Table 3 and Table 4.

*Table 3.* **Common hyperparameters.**

| Hyperparameter | Value |
|---|---|
| Learning rate | 0.0003 (default), 0.0001 (pointmaze) |
| Optimizer | Adam (Kingma & Ba, 2015) |
| # of gradient steps | 1000000 |
| Minibatch size | 1024 |
| Value network dimensions | $(512, 512, 512)$ |
| Policy network dimensions | $(512, 512, 512)$ (maze), $(256, 256)$ (robotic manipulation) |
| Nonlinearity | GELU (Hendrycks, 2016) |
| Target smoothing coefficient | 0.005 |
| Expectile $\kappa$ | 0.9 (maze), 0.7 (robotic manipulation) |
| LAN latent dimension | 64 |
| Subgoal representation dimension | 10 |
| Policy $(p^{\mathcal{D}}_{\text{cur}}, p^{\mathcal{D}}_{\text{traj}}, p^{\mathcal{D}}_{\text{geom}}, p^{\mathcal{D}}_{\text{rand}})$ ratio for $p^{\mathcal{D}}_{\text{mixed}}$ | $(0, 0.5, 0, 0.5)$ |
| Value $(p^{\mathcal{D}}_{\text{cur}}, p^{\mathcal{D}}_{\text{traj}}, p^{\mathcal{D}}_{\text{geom}}, p^{\mathcal{D}}_{\text{rand}})$ ratio for $p^{\mathcal{D}}_{\text{mixed}}$ | $(0.2, 0, 0.5, 0.3)$ (default), $(0.2, 0.3, 0, 0.5)$ ({antmaze, humanoidmaze}-stitch) |

*Table 4.* **Task-specific hyperparameters.**

| Environment | Dataset Type | LAVL Hyperparameters | | | | |
|---|---|---|---|---|---|---|
| | | $\gamma$ | $w_c$ | $\beta^h$ | $\beta^l$ | $k$ |
| Pointmaze | medium-navigate-v0 | 0.999 | 10.0 | 2.0 | 3.0 | 25 |
| | large-navigate-v0 | 0.999 | 10.0 | 1.0 | 3.0 | 25 |
| | giant-navigate-v0 | 0.999 | 10.0 | 1.0 | 3.0 | 25 |
| | medium-stitch-v0 | 0.999 | 10.0 | 0.5 | 3.0 | 25 |
| | large-stitch-v0 | 0.999 | 10.0 | 0.5 | 3.0 | 25 |
| | giant-stitch-v0 | 0.999 | 10.0 | 0.5 | 3.0 | 25 |
| Antmaze | medium-navigate-v0 | 0.999 | 10.0 | 1.0 | 3.0 | 25 |
| | large-navigate-v0 | 0.999 | 10.0 | 1.0 | 3.0 | 25 |
| | giant-navigate-v0 | 0.999 | 10.0 | 1.0 | 3.0 | 25 |
| | medium-stitch-v0 | 0.999 | 10.0 | 1.0 | 3.0 | 25 |
| | large-stitch-v0 | 0.999 | 10.0 | 1.0 | 3.0 | 25 |
| | giant-stitch-v0 | 0.999 | 10.0 | 1.0 | 3.0 | 25 |
| Humanoidmaze | medium-navigate-v0 | 0.999 | 0.0 | 2.0 | 3.0 | 100 |
| | large-navigate-v0 | 0.999 | 0.0 | 1.0 | 3.0 | 100 |
| | giant-navigate-v0 | 0.999 | 0.0 | 1.0 | 3.0 | 100 |
| | medium-stitch-v0 | 0.999 | 0.0 | 2.0 | 3.0 | 100 |
| | large-stitch-v0 | 0.999 | 0.0 | 1.0 | 3.0 | 100 |
| | giant-stitch-v0 | 0.999 | 0.0 | 1.0 | 3.0 | 100 |
| Cube | single-play-v0 | 0.99 | 0.0 | 0.5 | 10.0 | 10 |
| | double-play-v0 | 0.99 | 0.0 | 1.0 | 10.0 | 10 |
| | triple-play-v0 | 0.99 | 0.0 | 0.5 | 10.0 | 10 |
| Scene | play-v0 | 0.998 | 10.0 | 1.0 | 3.0 | 10 |

# B. Additional Experimental Results

## B.1. Comparison with CGCIVL

*Table 5.* **Comparison with CGCIVL.** We report the average (binary) success rate (%) across the five test-time goals on each task, averaged over five random seeds.

| Environment | Dataset Type | CGCIVL (official) | CGCIVL (reproduced) | LAVL (ours) |
|---|---|---|---|---|
| Pointmaze | medium-navigate-v0 | 87 ±4 | 76 ±4 | 92 ±1 |
| | large-navigate-v0 | 92 ±4 | 77 ±10 | 93 ±3 |
| | giant-navigate-v0 | 80 ±12 | 65 ±10 | 91 ±4 |
| | medium-stitch-v0 | 89 ±8 | 66 ±5 | 90 ±2 |
| | large-stitch-v0 | 98 ±2 | 79 ±3 | 87 ±8 |
| | giant-stitch-v0 | 81 ±17 | 47 ±19 | 95 ±2 |
| Antmaze | medium-navigate-v0 | 95 ±1 | 94 ±1 | 97 ±1 |
| | large-navigate-v0 | 91 ±2 | 90 ±1 | 93 ±1 |
| | giant-navigate-v0 | 73 ±5 | 65 ±2 | 85 ±2 |
| | medium-stitch-v0 | 91 ±3 | 88 ±2 | 97 ±1 |
| | large-stitch-v0 | 79 ±3 | 81 ±2 | 92 ±1 |
| | giant-stitch-v0 | 36 ±7 | 8 ±3 | 82 ±2 |
| Humanoidmaze | medium-navigate-v0 | 91 ±3 | 88 ±1 | 94 ±1 |
| | large-navigate-v0 | 58 ±8 | 46 ±2 | 74 ±3 |
| | giant-navigate-v0 | 29 ±9 | 10 ±2 | 83 ±2 |
| | medium-stitch-v0 | 90 ±2 | 89 ±1 | 93 ±2 |
| | large-stitch-v0 | 32 ±4 | 20 ±2 | 72 ±3 |
| | giant-stitch-v0 | 34 ±6 | 15 ±4 | 80 ±1 |
| Cube | single-play-v0 | 84 ±4 | 23 ±4 | 83 ±4 |
| | double-play-v0 | 46 ±4 | 7 ±2 | 42 ±8 |
| | triple-play-v0 | 5 ±2 | 5 ±1 | 11 ±2 |
| Scene | play-v0 | 77 ±5 | 56 ±16 | 88 ±4 |

CGCIVL (Ke et al., 2025) is a hierarchical GCRL algorithm that aims to mitigate the value inconsistency via conservative value estimation. While it achieves remarkable performance in OGBench datasets, CGCIVL features a critical implementation detail. The official implementation of CGCIVL includes (i) an oracle representation mode that utilizes pre-defined features (e.g., the x-y coordinate of maze environments) instead of learning subgoal representation, and (ii) a flat policy mode that extracts a flat policy instead of hierarchical two-level policies. We found that the two options have a significant impact on performance on some tasks. Specifically, the oracle representation greatly improves performance on Pointmaze tasks by circumventing the subgoal representation learning. Additionally, as evidenced by the contrast between GCIVL and HIQL in Table 1, flat policies outperform hierarchical policies on Cube tasks, and this tendency also applies to CGCIVL.

Although these implementation details are informative for practical purposes, applying different policy extraction methods for each task might mislead the comparison of algorithms. Therefore, we reproduce the benchmark evaluation results of CGCIVL in Table 5, using the same set of hyperparameters suggested by Ke et al. (2025), but applying hierarchical policy and learned subgoal representation for all tasks. The reproduced results are presented in Table 1 for comparison with baseline algorithms and LAVL.

## B.2. Comparison with Eik-HIQL

*Table 6.* **Comparison with Eik-HIQL and Eik-HiQRL.** We report the average success rate (%) across the five test-time goals on each task, averaged over eight random seeds. To match the evaluation protocol of Giammarino et al. (2025) and Giammarino & Qureshi (2025), we report the *best* evaluation during the training.

| Environment | Dataset Type | Eik-HIQL | Eik-HiQRL | LAVL (ours) |
|---|---|---|---|---|
| Pointmaze | `medium-navigate-v0` | $93_{\pm5}$ | $\mathbf{99}_{\pm1}$ | $97_{\pm2}$ |
| | `large-navigate-v0` | $83_{\pm9}$ | $\mathbf{99}_{\pm1}$ | $\mathbf{99}_{\pm1}$ |
| | `giant-navigate-v0` | $79_{\pm13}$ | $89_{\pm8}$ | $\mathbf{97}_{\pm1}$ |
| | `medium-stitch-v0` | $96_{\pm3}$ | $\mathbf{99}_{\pm2}$ | $98_{\pm1}$ |
| | `large-stitch-v0` | $73_{\pm6}$ | $95_{\pm8}$ | $\mathbf{97}_{\pm2}$ |
| | `giant-stitch-v0` | $22_{\pm10}$ | $57_{\pm14}$ | $\mathbf{98}_{\pm1}$ |
| Antmaze | `medium-navigate-v0` | $95_{\pm1}$ | $95_{\pm2}$ | $\mathbf{98}_{\pm1}$ |
| | `large-navigate-v0` | $86_{\pm2}$ | $86_{\pm2}$ | $\mathbf{94}_{\pm1}$ |
| | `giant-navigate-v0` | $67_{\pm5}$ | $66_{\pm1}$ | $\mathbf{88}_{\pm3}$ |
| | `medium-stitch-v0` | $94_{\pm2}$ | $94_{\pm2}$ | $\mathbf{97}_{\pm1}$ |
| | `large-stitch-v0` | $84_{\pm3}$ | $88_{\pm3}$ | $\mathbf{94}_{\pm1}$ |
| | `giant-stitch-v0` | $48_{\pm11}$ | $61_{\pm9}$ | $\mathbf{85}_{\pm1}$ |
| Humanoidmaze | `medium-navigate-v0` | $86_{\pm2}$ | $91_{\pm2}$ | $\mathbf{95}_{\pm1}$ |
| | `large-navigate-v0` | $64_{\pm7}$ | $74_{\pm5}$ | $\mathbf{78}_{\pm3}$ |
| | `giant-navigate-v0` | $68_{\pm5}$ | $83_{\pm6}$ | $\mathbf{86}_{\pm2}$ |
| | `medium-stitch-v0` | $79_{\pm2}$ | $85_{\pm3}$ | $\mathbf{95}_{\pm2}$ |
| | `large-stitch-v0` | $29_{\pm7}$ | $63_{\pm6}$ | $\mathbf{76}_{\pm4}$ |
| | `giant-stitch-v0` | $19_{\pm5}$ | $69_{\pm5}$ | $\mathbf{85}_{\pm2}$ |
| Cube | `single-play-v0` | $25_{\pm1}$ | $12_{\pm3}$ | $\mathbf{85}_{\pm3}$ |
| Scene | `play-v0` | $52_{\pm7}$ | $55_{\pm14}$ | $\mathbf{91}_{\pm3}$ |

Eik-HIQL (Giammarino et al., 2025) and Eik-HiQRL (Giammarino & Qureshi, 2025) are hierarchical GCRL algorithms inspired by the Eikonal partial differential equation (Noack & Clark, 2017). The principle is implemented by regularization or constraint on $\nabla_s V(s, g)$, which induces a smooth value surface. They demonstrated the efficiency of the gradient-based approach in OGBench datasets. However, their evaluation protocol differs from the standard of OGBench, where the former measures the best evaluation during the training, while the latter takes the average of the last three evaluations during the training. Therefore, for fair comparison, we present the comparison with Eik-HIQL and Eik-HiQRL in Table 6, where we report the results under the same evaluation protocol as them.

## B.3. Optimization Stability of LAN

In this section, we analyze the optimization stability of LAN, which constitutes one of its key empirical advantages. We begin by introducing a proxy measure for assessing the quality of the learned value function.

**Proxy Measure for Value Quality.** To measure the quality of learned value function, we introduce *Kendall order consistency*, a metric inspired by Kendall's Tau (Kendall, 1938) and the order consistency ratio from Ahn et al. (2025).

**Definition B.1** (Kendall Order Consistency). Let $\tau^\star = (s_0, s_1, \ldots, s_T = g)$ be an optimal trajectory from an initial state $s_0$ to the goal $g$, and let $V$ be a goal-conditioned value function. We define the Kendall order consistency of $V$ evaluated on $\tau^\star$ as

$$\mathcal{K}(V, \tau^\star) := \frac{2}{T(T+1)} \sum_{0 \le i < j \le T} \mathbb{1}\left[V(s_j, g) > V(s_i, g)\right],$$

where $\mathbb{1}[\cdot]$ denotes the indicator function. Therefore, the Kendall order consistency measures how well $V$ is aligned with the optimal trajectory $\tau^\star$.

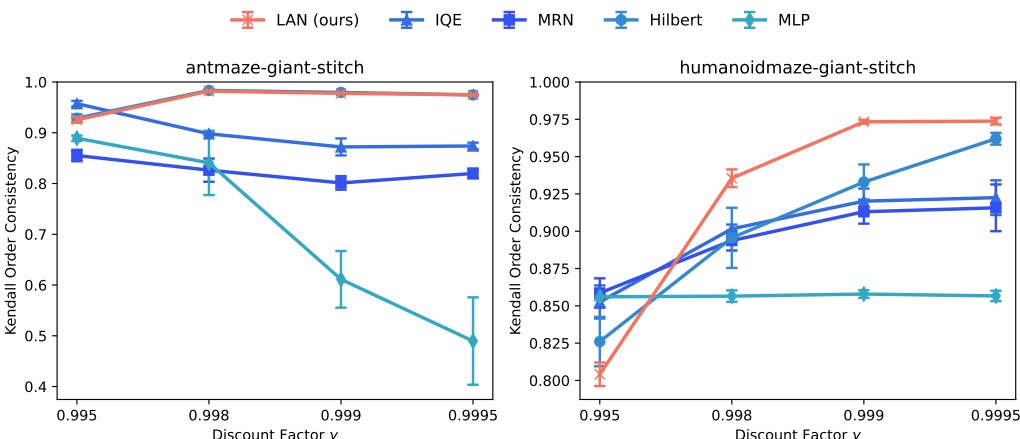

*Figure 8.* Kendall order consistency of GCIVL agents with different value function architectures (LAN, IQE, MRN, Hilbert, and MLP).

Note that Kendall order consistency serves as a proxy rather than a direct predictor of task success, since capturing monotonicity along a particular trajectory does not guarantee accurate value estimates in the neighbor state space. In other words, the high order consistency is necessary but not sufficient for strong performance.

To compute the Kendall order consistency, we use the optimal trajectory datasets provided by Ahn et al. (2025), which are collected by the expert policies originally used to generate OGBench datasets.

**Order Consistency Analysis.**

In long-horizon GCRL, TD-based value learning becomes increasingly unstable due to sparse reward signals. Accumulated estimation errors can lead to oscillatory behavior or even divergence of value estimates, making stable convergence difficult for methods such as GCIVL. Figure 8 illustrates this issue by comparing the Kendall order consistency of GCIVL agents with different value function architectures.

As the discount factor increases, MLP-parameterized value functions rapidly lose order consistency, in some cases approaching a Kendall score of $0.5$, which corresponds to random ordering. Quasimetric architectures exhibit greater robustness but still suffer noticeable degradation in challenging environments such as `antmaze-giant-stitch`, though they do not completely collapse.

These observations help explain the poor performance of vanilla GCIVL in long-horizon tasks. Environments such as `giant` mazes often require hundreds or thousands of steps to reach the goal (e.g., the horizon of `humanoidmaze-giant-stitch` in OGBench is 4000). Consequently, discount factors exceeding $0.995$ are necessary to propagate sparse terminal rewards throughout the state space. However, in this regime, MLP-based value functions exhibit severe instability and overgeneralization, frequently resulting in training failure or erroneous policies.

In contrast, LAN maintains high Kendall order consistency even when the discount factor is close to $1$. While metric-based (Hilbert) parameterizations achieve comparable performance to LAN in `antmaze-giant-stitch`, LAN substantially outperforms them in `humanoidmaze-giant-stitch`. These results indicate that LAN offers superior stability and consistency in the long-horizon regime, highlighting a key empirical advantage of the proposed architecture.

## B.4. Additional Experiments on Value Generalization

We present additional experimental results supporting our finding that value generalization is a fundamental bottleneck of offline GCRL, and inductive bias is the key to alleviating value overgeneralization.

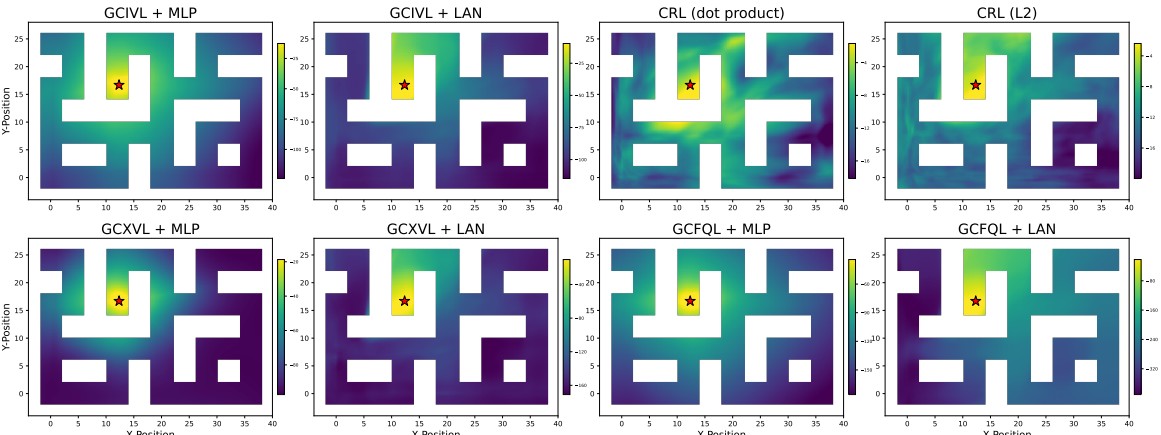

*Figure 9.* Comparison of learned value function of offline GCRL algorithms on a maze navigation task (`antmaze-large-stitch`). For CRL, similarity functions are visualized.

**Generalization issue in offline GCRL.**    To demonstrate that our findings on value overgeneralization is not restricted to GCIVL, we conduct additional experiments visualizing the value landscape learned by other offline GCRL algorithms: (1) CRL, a contrastive learning algorithm for goal-conditioned value learning, (2) GCFQL, a goal-conditioned extension of FQL (Park et al., 2025b) that uses a flow-matching policy within an actor-critic framework, (3) GCXVL, a goal-conditioned variant of XQL that learns a goal-conditioned value function via Gumbel regression for MaxEnt RL. Since CRL admits multiple choices of similarity function, we consider the dot-product similarity used in the original CRL paper (Eysenbach et al., 2022) and the L2-distance-based similarity used in follow-up works such as Wang et al. (2026). For each algorithm, we follow the default hyperparameters suggested by OGBench or official implementation.

Figure 9 visualizes the learned value functions of these methods (and the learned similarity function in the case of CRL). The results suggest that generalization issue arises across multiple offline GCRL algorithms:

- CRL exhibits irregular errors throughout the state space. Its learned landscape contains substantial local fluctuations ("ripples"), indicating that the learned similarity function does not form a smooth geometry aligned with the environment dynamics. This appears to stem from the contrastive nature of the CRL objective. These results suggest that, while overgeneralization is a prominent and recurring failure mode, it is not the only way in which learned goal-conditioned geometry can become misaligned with temporal structure.

- With dot-product similarity, CRL clearly exhibits overgeneralization: states that are close to the goal in Euclidean distance receive high similarity even when they are far in temporal distance. In contrast, with L2-based similarity, CRL shows little visible overgeneralization apart from the irregular local errors described above.

- GCFQL and GCXVL show a pattern consistent with GCIVL. MLP-parameterized value functions exhibit overgeneralization, whereas LAN yields much more accurate generalization. This provides additional evidence that the inductive bias of LAN can be beneficial beyond GCIVL, and may extend to other offline GCRL algorithms as well.

Overall, these additional results strengthen our diagnosis that reliable offline GCRL depends critically on accurate generalization that is aligned with temporal distance.

**Effect of inductive bias for TD learning.**    To verify whether architectural inductive bias in value function improves value learning methods other than GCIVL, we compare three value function architectures under the GCXVL loss. Table 7 shows that GCXVL with LAN tends to match or outperform GCXVL with MLP and GCXVL with IQE. The results indicate that the findings of Section 6 extend beyond GCIVL to GCXVL.

*Table 7.* Performance of GCXVL agents with three different value function parameterization. We consider the temperature of GCXVL $\beta_{XVL} \in \{1.0, 3.0, 10.0\}$. We report the average (binary) success rate (%) across the five test-time goals on each task, averaged over 3 random seeds.

| Dataset Type | $\beta_{XVL}$ | GCXVL + MLP | GCXVL + IQE | GCXVL + LAN |
|---|---|---|---|---|
| Antmaze-medium-navigate-v0 | 1 | $49_{\pm 6}$ | $62_{\pm 5}$ | $73_{\pm 5}$ |
| | 3 | $40_{\pm 16}$ | $90_{\pm 1}$ | $89_{\pm 3}$ |
| | 10 | $85_{\pm 2}$ | $90_{\pm 1}$ | $89_{\pm 2}$ |
| Antmaze-large-navigate-v0 | 1 | $18_{\pm 3}$ | $30_{\pm 5}$ | $32_{\pm 16}$ |
| | 3 | $17_{\pm 7}$ | $44_{\pm 9}$ | $54_{\pm 2}$ |
| | 10 | $41_{\pm 1}$ | $43_{\pm 4}$ | $52_{\pm 1}$ |
| Cube-single-play-v0 | 1 | $24_{\pm 7}$ | $23_{\pm 2}$ | $5_{\pm 1}$ |
| | 3 | $47_{\pm 4}$ | $28_{\pm 4}$ | $48_{\pm 5}$ |
| | 10 | $32_{\pm 3}$ | $26_{\pm 2}$ | $54_{\pm 4}$ |
| Cube-double-play-v0 | 1 | $5_{\pm 0}$ | $33_{\pm 10}$ | $1_{\pm 0}$ |
| | 3 | $37_{\pm 6}$ | $40_{\pm 3}$ | $51_{\pm 1}$ |
| | 10 | $21_{\pm 2}$ | $31_{\pm 4}$ | $48_{\pm 1}$ |

## B.5. Additional Value Plots

We present the extended value plots from the experiment discussed in Section 4, including the results with the MRN, metric parameterization (Hilbert), and our LAN architecture. The results prove that inductive bias, not the quasimetric property, is the key to alleviating value overgeneralization. Since the `antmaze-large-stitch` dataset provides five evaluation tasks (state-goal pairs) where the fourth and fifth goals are identical, Figure 10–13 shows the value plot for task 1–4.

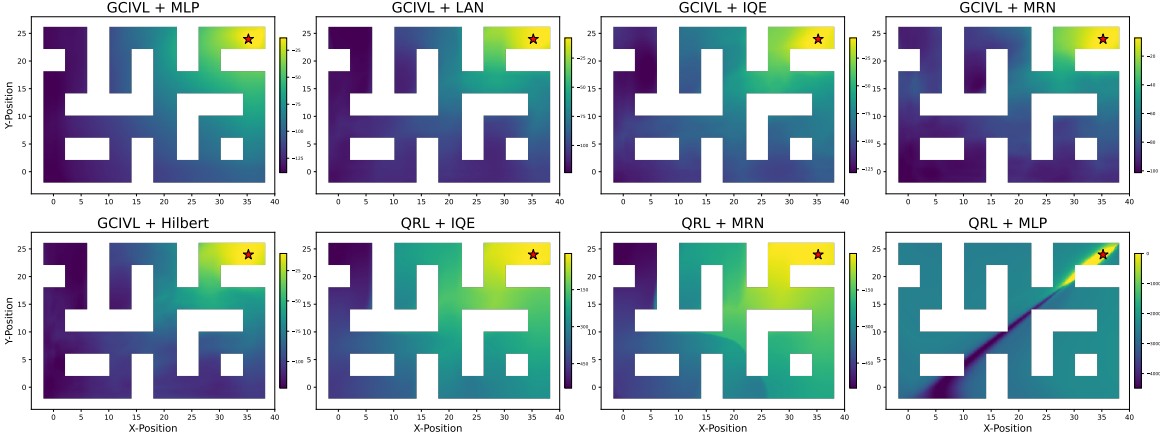

*Figure 10.* Full value plots for task 1 of `antmaze-large-stitch`

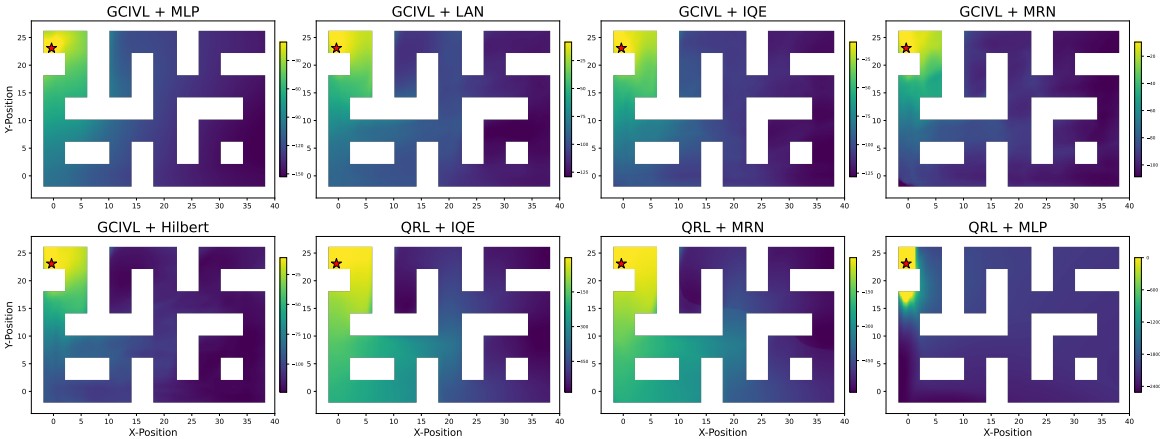

*Figure 11.* Full value plots for task 2 of `antmaze-large-stitch`

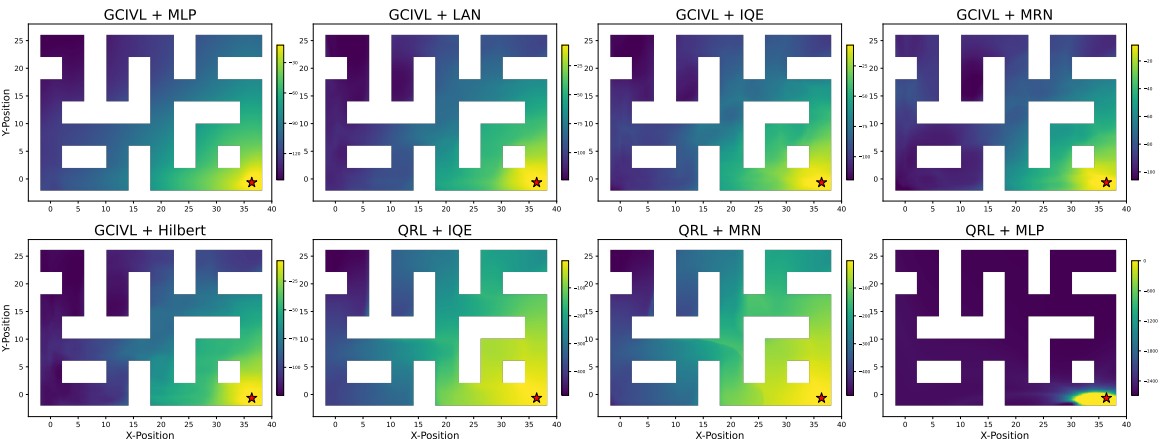

*Figure 12.* Full value plots for task 3 of `antmaze-large-stitch`

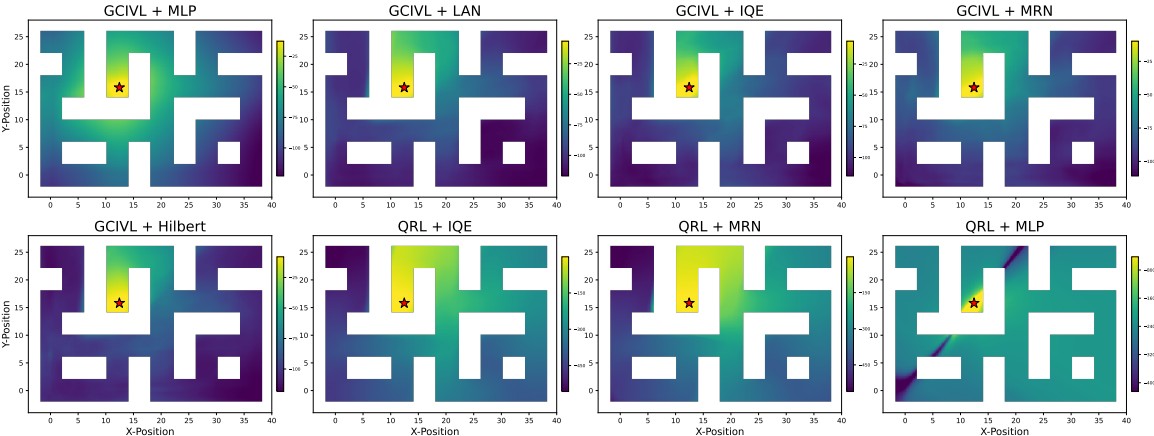

*Figure 13.* Full value plots for task 4 of `antmaze-large-stitch`

## B.6. More Ablation Experiments

**Alternative value parameterization.** While we compare LAN with existing value parameterization methods in Section 5 and Section 6, we further compare alternative value parameterizaiton methods: Dot Product ($= \varphi_S(s)^T \varphi_G(g)$), Bilinear ($= \varphi_S(s)^T W \varphi_G(g)$), and Learned Similarity ($= f([\varphi_S(s), \varphi_G(g)])$) where $f(\cdot)$ is a learnable MLP.

In Table 8, LAN consistently outperforms the alternative designs across datasets. Dot-product and bilinear parameterizations remain competitive in some tasks but degrade in more challenging settings, while learned similarity performs poorly overall. This supports our claim that effective value learning depends not only on expressivity, but also on appropriate inductive bias.

*Table 8.* Ablation study on the value parameterization within LAVL. We compare LAN with four asymmetric value parameterization methods. We set $\beta^h = 1.0$ for all runs, and report the average (binary) success rate (%) across the five test-time goals on each task, averaged over 3 random seeds.

| Dataset Type | LAN (ours) | Dot Product | Bilinear | Learned Similarity |
|---|---|---|---|---|
| Pointmaze-giant-stitch-v0 | **91** $_{\pm 7}$ | 67 $_{\pm 4}$ | 73 $_{\pm 3}$ | 0 $_{\pm 0}$ |
| Antmaze-giant-stitch-v0 | **78** $_{\pm 2}$ | 75 $_{\pm 2}$ | 77 $_{\pm 1}$ | 34 $_{\pm 1}$ |
| Humanoidmaze-giant-stitch-v0 | **80** $_{\pm 1}$ | 0 $_{\pm 0}$ | 5 $_{\pm 3}$ | 6 $_{\pm 1}$ |
| Cube-single-play-v0 | **79** $_{\pm 4}$ | 64 $_{\pm 3}$ | 66 $_{\pm 4}$ | 40 $_{\pm 2}$ |
| Scene-play-v0 | **87** $_{\pm 6}$ | 26 $_{\pm 6}$ | 31 $_{\pm 4}$ | 5 $_{\pm 1}$ |

**Effect of LAN-based value learning and hierarchical policy.** To disentangle the contributions of value learning and hierarchy in LAVL, we ablate LAN-based value learning and the hierarchical policy. Removing LAN reduces LAVL to HIQL, replacing the hierarchical policy under LAN yields GCIVL+LAN, and removing both reduces to GCIVL.

Table 9 clearly shows that LAN improves performance under both flat and hierarchical policies, highlighting the importance of accurate value generalization in GCRL. Furthermore, the hierarchy consistently helps with navigation, but on manipulation tasks, it only improves LAN-parameterized methods. This suggests that hierarchical policies are most effective when combined with LAN.

*Table 9.* Ablation study on LAN-based value learning and the hierarchical policy. We set $\beta^h = 1.0$ for all runs, and report the average (binary) success rate (%) across the five test-time goals on each task, averaged over 3 random seeds.

| Dataset Type | GCIVL | GCIVL + LAN | HIQL | LAVL (ours) |
|---|---|---|---|---|
| Antmaze-medium-navigate-v0 | 72 $_{\pm 8}$ | 88 $_{\pm 1}$ | 96 $_{\pm 1}$ | **97** $_{\pm 1}$ |
| Antmaze-large-navigate-v0 | 16 $_{\pm 5}$ | 52 $_{\pm 3}$ | 91 $_{\pm 2}$ | **92** $_{\pm 1}$ |
| Antmaze-giant-navigate-v0 | 0 $_{\pm 0}$ | 1 $_{\pm 0}$ | 65 $_{\pm 5}$ | **83** $_{\pm 1}$ |
| Cube-single-navigate-v0 | 53 $_{\pm 4}$ | 68 $_{\pm 2}$ | 15 $_{\pm 3}$ | **79** $_{\pm 4}$ |
| Scene-navigate-v0 | 42 $_{\pm 4}$ | 64 $_{\pm 5}$ | 38 $_{\pm 3}$ | **87** $_{\pm 6}$ |

**Value Architecture and Continuity Regularization.** The performance gains of LAVL over the baseline stem primarily from the LAN-based value parameterization. Continuity regularization addresses residual local errors after LAN mitigates overgeneralization. To disentangle the contributions of LAN and continuity regularization, we compare four configurations that vary (i) the value parameterization (LAN vs. MLP) and (ii) whether continuity regularization is used. Since replacing LAN with MLP in LAVL corresponds to HIQL, we use the HIQL implementation for that variant.

Table 10 shows that the largest improvement comes from replacing the MLP value function with LAN. Continuity regularization provides little benefit in HIQL but yields a further gain when combined with LAN. This suggests that LAN addresses the dominant bottleneck, while continuity regularization helps with remaining local errors.

**Continuity regularization for alternative value architectures.** To verify that continuity regularization also benefits other value architectures, we replaced LAN in LAVL with IQE, MRN, and Hilbert and evaluated the resulting variants under different regularization strengths.

Table 11 shows that continuity regularization improves average performance across architectures, although the gains are

*Table 10.* Ablation study on value architectures (LAN vs. MLP) and the continuity regularization. For HIQL, we adopt the hyperparameters suggested in OGBench. For LAVL, we set $\beta^h = 1.0$ across all runs for a fair comparison. Additionally, when using continuity regularization, we searched over $w_c \in \{1, 10\}$ for both HIQL and LAVL. We report the average (binary) success rate (%) across the five test-time goals on each task, averaged over 3 random seeds.

| Dataset Type | HIQL | HIQL with Reg. | LAVL without Reg. | LAVL with Reg. |
|---|---|---|---|---|
| Pointmaze-giant-stitch-v0 | $0_{\pm 0}$ | $4_{\pm 3}$ | $28_{\pm 10}$ | $\mathbf{91}_{\pm 7}$ |
| Antmaze-giant-stitch-v0 | $2_{\pm 2}$ | $2_{\pm 2}$ | $68_{\pm 3}$ | $\mathbf{78}_{\pm 2}$ |
| Humanoidmaze-giant-stitch-v0 | $3_{\pm 2}$ | $3_{\pm 2}$ | $\mathbf{80}_{\pm 1}$ | $77_{\pm 1}$ |
| Cube-single-play-v0 | $15_{\pm 3}$ | $19_{\pm 5}$ | $\mathbf{79}_{\pm 4}$ | $66_{\pm 1}$ |
| Scene-play-v0 | $69_{\pm 7}$ | $69_{\pm 7}$ | $69_{\pm 7}$ | $\mathbf{87}_{\pm 6}$ |

dataset-dependent, so we treat $w_c$ as a hyperparameter. LAN performs best at every value of $w_c$, indicating that its advantage is orthogonal to the effect of continuity regularization.

*Table 11.* Ablation study on continuity regularization across different value function architectures. Based on LAVL, we consider the continuity regularization hyperparameter $w_c \in \{0, 1.0, 10.0\}$ for LAN, IQE, MRN, and Hilbert. We set $\beta^h = 1.0$ for all runs, and report the average (binary) success rate (%) across the five test-time goals on each task, averaged over 3 random seeds.

| Dataset Type | $w_c$ | LAN (ours) | IQE | MRN | Hilbert |
|---|---|---|---|---|---|
| Pointmaze-giant-stitch-v0 | 0 | $28_{\pm 10}$ | $25_{\pm 1}$ | $29_{\pm 1}$ | $36_{\pm 5}$ |
| | 1 | $56_{\pm 9}$ | $45_{\pm 11}$ | $27_{\pm 2}$ | $36_{\pm 12}$ |
| | 10 | $91_{\pm 7}$ | $39_{\pm 3}$ | $20_{\pm 6}$ | $46_{\pm 9}$ |
| Antmaze-giant-stitch-v0 | 0 | $68_{\pm 3}$ | $34_{\pm 20}$ | $22_{\pm 13}$ | $65_{\pm 3}$ |
| | 1 | $76_{\pm 2}$ | $78_{\pm 2}$ | $71_{\pm 1}$ | $70_{\pm 2}$ |
| | 10 | $78_{\pm 2}$ | $78_{\pm 3}$ | $73_{\pm 6}$ | $77_{\pm 1}$ |
| Humanoidmaze-giant-stitch-v0 | 0 | $80_{\pm 1}$ | $44_{\pm 1}$ | $51_{\pm 8}$ | $82_{\pm 4}$ |
| | 1 | $77_{\pm 1}$ | $80_{\pm 3}$ | $75_{\pm 5}$ | $83_{\pm 2}$ |
| | 10 | $57_{\pm 3}$ | $78_{\pm 1}$ | $75_{\pm 3}$ | $80_{\pm 2}$ |
| Cube-single-play-v0 | 0 | $79_{\pm 4}$ | $31_{\pm 9}$ | $35_{\pm 4}$ | $39_{\pm 3}$ |
| | 1 | $66_{\pm 1}$ | $27_{\pm 3}$ | $35_{\pm 2}$ | $40_{\pm 2}$ |
| | 10 | $66_{\pm 8}$ | $34_{\pm 3}$ | $33_{\pm 4}$ | $33_{\pm 3}$ |
| Scene-play-v0 | 0 | $69_{\pm 7}$ | $26_{\pm 6}$ | $31_{\pm 4}$ | $5_{\pm 1}$ |
| | 1 | $66_{\pm 6}$ | $15_{\pm 3}$ | $14_{\pm 3}$ | $9_{\pm 3}$ |
| | 10 | $87_{\pm 6}$ | $20_{\pm 4}$ | $13_{\pm 2}$ | $14_{\pm 2}$ |

## B.7. Hyperparameter Sensitivity

**Performance with minimal tuning.** Most hyperparameters of LAVL were inherited from HIQL and the default values of OGBench, as discussed in Appendix A.3. Tuning parameters were selected in a principled manner:

- $\gamma \in \{0.99, 0.998, 0.999\}$, $\kappa \in \{0.7, 0.9\}$, and $\beta^l \in \{3, 10\}$ were tuned for each environment (maze, cube, scene).

- $w_c \in \{0, 1, 10\}$ was tuned for each task ({point, ant, humanoid}maze, cube, scene).

- Only $\beta^h \in \{0.5, 1, 2\}$ was tuned per dataset.

To further verify hyperparameter sensitivity, we evaluate a minimal-tuning variant of LAVL with the temperature parameters and $\kappa$ fixed to their default values, $(\beta^h, \beta^l) = (1, 3)$ and $\kappa = 0.7$. Table 12 shows that the minimally tuned LAVL still maintains high success rates on most datasets, verifying that LAVL does not rely on heavy dataset-specific tuning.

*Table 12.* Performance of LAVL with minimal hyperparameter tuning where the temperature parameters and $\kappa$ fixed to their default values, $(\beta^h, \beta^l) = (1, 3)$ and $\kappa = 0.7$, across all datasets. Because LAVL (minimal tuning) was evaluated over three seeds, part of the observed performance improvement may be attributable to seed-level randomness.

| Environment | Dataset Type | LAVL (minimal tuning) | LAVL (tuned) |
|---|---|---|---|
| Pointmaze | medium-navigate-v0 | 91 ±2 | 92 ±1 |
| | large-navigate-v0 | 97 ±2 | 93 ±3 |
| | giant-navigate-v0 | 86 ±2 | 91 ±4 |
| | medium-stitch-v0 | 80 ±6 | 90 ±2 |
| | large-stitch-v0 | 91 ±5 | 87 ±8 |
| | giant-stitch-v0 | 90 ±3 | 95 ±2 |
| Antmaze | medium-navigate-v0 | 98 ±1 | 97 ±1 |
| | large-navigate-v0 | 93 ±1 | 93 ±1 |
| | giant-navigate-v0 | 82 ±1 | 85 ±2 |
| | medium-stitch-v0 | 94 ±1 | 97 ±1 |
| | large-stitch-v0 | 90 ±1 | 92 ±1 |
| | giant-stitch-v0 | 72 ±1 | 82 ±2 |
| Humanoidmaze | medium-navigate-v0 | 91 ±1 | 94 ±1 |
| | large-navigate-v0 | 70 ±1 | 74 ±3 |
| | giant-navigate-v0 | 75 ±5 | 83 ±2 |
| | medium-stitch-v0 | 90 ±2 | 93 ±2 |
| | large-stitch-v0 | 53 ±7 | 72 ±3 |
| | giant-stitch-v0 | 71 ±7 | 80 ±1 |
| Cube | single-play-v0 | 79 ±4 | 83 ±4 |
| | double-play-v0 | 39 ±5 | 42 ±8 |
| | triple-play-v0 | 12 ±5 | 11 ±2 |
| Scene | play-v0 | 87 ±6 | 88 ±4 |

**Bandit-based analysis for hyperparameter sensitivity.** While we provide detailed hyperparameter values (Appendix A.3) and the hyperparameter robustness of LAVL (Table 12), we further conduct bandit-based sensitivity analysis suggested in Jackson et al. (2026). We first collect a pool of online evaluation scores from policies trained by LAVL and HIQL. Concretely, we train 24 candidate policies for each method from fixed hyperparameter ranges and random seeds:

- LAVL: we vary the high-level temperature $\beta^h \in \{0.5, 1, 2, 3\}$ and the continuity regularization coefficient $w_c \in \{0, 1, 10\}$, and train 2 random seeds for each configuration.

- HIQL: we vary $\beta^h \in \{0.5, 1, 2, 3\}$ and train 6 random seeds per value.

For other parameters, such as the goal-sampling ratio and the low-level temperature, we use the same values for both methods. Each policy is evaluated online, and the resulting scores are used for the subsequent bandit evaluation.

Following the spirit of the bandit-based evaluation in Jackson et al. (2026), we perform policy selection over a fixed pool of candidate policies under different tuning budgets. Since our experiments are conducted on OGBench, where performance is summarized by success rate rather than episodic return, we use one policy-level score per candidate policy, obtained from 50 evaluation episodes and averaged over 5 goals. Specifically, we subsample 8 candidate policies and run a UCB bandit over them. We evaluate performance as a function of the tuning budget, up to a maximum of 200 policy evaluations. For each budget, results are averaged over 500 repeated bandit rollouts, and confidence intervals are estimated using 2000 bootstrap samples.

To keep the environment selection consistent, we evaluate on six datasets: `antmaze-giant-navigate`, `antmaze-giant-stitch`, `humanoidmaze-large-navigate`, and `humanoidmaze-large-stitch` from the maze-navigation domain, and `cube-single-play` and `scene-play` from the robotic manipulation domain.

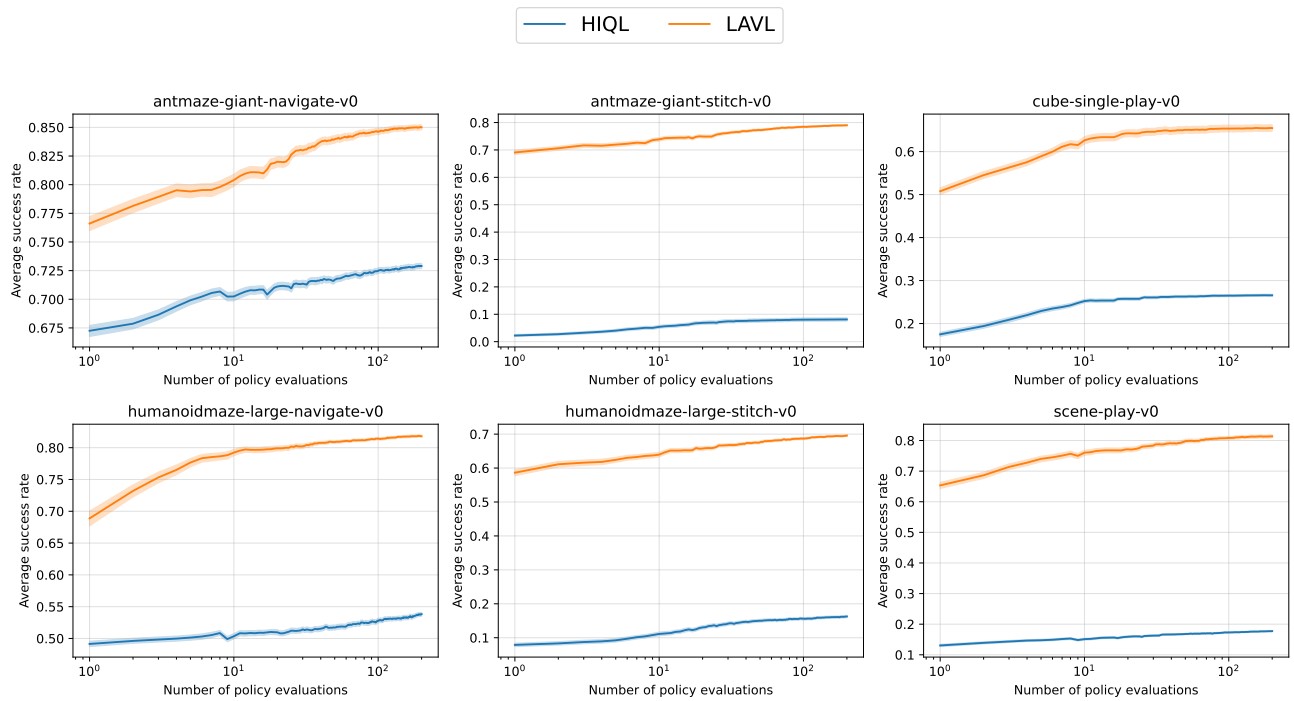

*Figure 14.* Bandit-based evaluation of HIQL and LAVL–mean and 95% confidence intervals over 500 bandit rollouts, with 8 policy arms subsampled from 24 trained policies in each rollout. The $x$-axis denotes the number of bandit pulls, while the $y$-axis denotes the average success rate of the arm estimated to be best after $x$ pulls.

In Figure 14, we present the bandit-based evaluation of HIQL and LAVL on six OGBench datasets. Generally, a method performs better if its curve is closer to the top-left corner of the plot, indicating higher success rates with fewer online policy evaluations.

Across all six datasets, LAVL consistently outperforms HIQL over the full range of tuning budgets. The advantage is especially pronounced on the stitch datasets, where HIQL remains far below LAVL even with larger policy evaluation budgets. This suggests that LAVL provides a stronger and more robust candidate policy pool, rather than benefiting only from favorable policy selection after extensive tuning. Importantly, the ordering between the two methods is stable across the budget range, suggesting that LAVL's improvement is not driven merely by aggressive tuning, but by better policy quality and robustness.

# C. Implementation Details

## C.1. Quasimetric Value Architectures

It is known that the optimal goal-conditioned value function $V^\star(s, g)$ satisfies the triangle inequality $V^\star(s_1, s_2) + V^\star(s_2, s_3) \leq V^\star(s_1, s_3)$ for arbitrary $s_1, s_2, s_3 \in \mathcal{S}$ (Wang et al., 2023). Therefore, $-V^\star$ is a quasimetric over $\mathcal{S}$. Building on this fact, some prior works utilized specialized quasimetric network architectures for value learning (Liu et al., 2023; Wang et al., 2023).

**Metric Residual Network (MRN)** (Liu et al., 2023) implements the quasimetric property as the sum of a symmetrical metric term and an asymmetrical residual term. Specifically, it formulates $V(s, g) = -\|\varphi(s) - \varphi(g)\|_2 - \max_{i \in [K]} \text{relu}(h_i(s) - h_i(g))$ where $\varphi : \mathcal{S} \to \mathbb{R}^d$, $h : \mathcal{S} \to \mathbb{R}^K$ are state encoders. For notational coherence, MRN can be written as $V(s, g) = -\tilde{d}(\varphi(s), \varphi(g))$ where $\tilde{d}(x, y) = \|x_{:d} - y_{:d}\|_2 + \max_{j \in [K]} \text{relu}(x_{j+d} - y_{j+d})$ for $x, y \in \mathbb{R}^{d+K}$.

**Interval Quasimetric Embedding (IQE)** (Wang & Isola, 2022) first embeds the state with $k$ encoders, $\{\varphi_i\}_{i \in [K]}$. For each $u_i = \varphi_i(s)$ and $v_i = \varphi_i(g)$, the similarity is measured by the Lebesgue measure of unions of several intervals:

$$d_i(u, v) = \left| \bigcup_{j=1}^{L} [u_{ij}, \max(u_{ij}, v_{ij})] \right| \text{ for } i \in [K],$$

where $u = [u_1, \ldots, u_K]^T, v = [v_1, \ldots, v_K]^T \in \mathbb{R}^{K \times L}$ Then, each component $d_i$ is combined to define the final quasimetric by simply summing the components as $d_{\text{IQE-sum}}(u, v) = \sum_{i=1}^{K} d_i(u, v)$, or using the maxmean reduction as $d_{\text{IQE-maxmean}}(u, v) = \alpha \max_{i \in [K]} d_i(u, v) + (1 - \alpha)\text{mean}_{i \in [K]} d_i(u, v)$. Following the implementation of OGBench, we use the maxmean reduction with learnable $\alpha$. Therefore, the value is parameterized as $V(s, g) = -\tilde{d}(\varphi(s), \varphi(g))$ where $\tilde{d} = d_{\text{IQE-maxmean}}$.

**Metric Parameterization (Hilbert Representation)** (Sermanet et al., 2018; Ma et al., 2023; Park et al., 2024) utilizes the metric structure $V(s, g) = -\|\varphi(s) - \varphi(g)\|_2$ to train a state encoder $\phi$, which is a stronger constraint than the quasimetric property. To compare with MRN and IQE, we can interpret it as $V(s, g) = -\tilde{d}(\varphi(s), \varphi(g))$ where $\tilde{d}(x, y) = \|x - y\|_2$. As we discussed in Section 5.1, the architecture is used to train a state encoder $\phi$ for downstream RL or zero-shot planning, and the value $-\|\varphi(s) - \varphi(g)\|_2$ is not used directly. However, we demonstrate that the inductive bias in the metric structure helps value learning in some tasks (Section 6.4), discovering a novel potential of the Hilbert representation.

## C.2. Hierarchical Policy Framework

The hierarchical policy framework of HIQL has been widely used for long-horizon GCRL tasks. However, as we discussed in Section 6.5, there are subtle details in the implementation of hierarchy. In this section, we present details on the hierarchical policy extraction.

**Training subgoal representation with policy loss.** Recall that the policy losses in Section 5.3 contain the subgoal representation network $\phi$:

$$\mathcal{L}(\pi^h) = \mathbb{E}_{(s_t, s_{t+k}) \in \mathcal{D}, g \sim p^{\mathcal{D}}_{\text{mixed}}(\cdot | s_t)} \left[ -e^{\beta^h A^h(s_t, s_{t+k}, g)} \log \pi^h(\phi(s_{t+k}) \mid s_t, g) \right],$$

$$\mathcal{L}(\pi^l) = \mathbb{E}_{(s_t, a_t, s_{t+1}, s_{t+k}) \in \mathcal{D}} \left[ -e^{\beta^l A^l(s_t, s_{t+1}, g)} \log \pi^l(a_t \mid s_t, \phi(s_{t+k})) \right].$$

Therefore, it is possible to train $\phi$ with the policy losses. Indeed, Park et al. (2023) pointed out that allowing the gradient of $\mathcal{L}(\pi^l)$ flow through $\phi$ improves performance in some tasks. Building on this, we found that additionally using the gradient of $\mathcal{L}(\pi^h)$ further improves performance in maze tasks. However, the gradient of policy loss is often large compared to that of value loss, due to the large scale of exponential advantage terms $\exp(\beta^h A^h(s_t, s_{t+k}, g))$ and $\exp(\beta^l A^l(s_t, s_{t+1}, g))$. To stabilize the training, we normalize the exponential advantage term by the batch mean. In our experiments, the gradient flow from the policy losses to the subgoal representation is activated for maze tasks, which slightly improved performance.

**Quasimetric architectures.** Now we elaborate on the implementation of LAVL variants in Section 6.4. As we discussed in Section C.1, the quasimetric architects take the form $V(s, g) = -\tilde{d}(\varphi(s), \varphi(g))$ for some state encoder $\varphi$. To apply these value architectures to the hierarchical framework, the inputs $(s, g)$ have to be replaced by corresponding subgoal representations $(\phi(s), \phi(g))$, which leads to $V(s, g) = -\tilde{d}(\varphi(\phi(s)), \varphi(\phi(g)))$. The composition $\varphi(\phi(s))$ looks redundant, but this is essential, since the dimension of the latent space is much larger than the dimension of the subgoal representation.

Under the default setting of OGBench implementation, the former is 512 while the latter is 10, which is two orders of magnitude smaller. Thus, we can interpret $\phi$ as a bottleneck representation for state encoding, expected to contain useful features for control. Equipped with the value parameterization, hierarchical policy extraction can be done in the same manner as LAVL.

**LAVL-HV.** As explained in Section 6.5, LAVL-HV is a midpoint of LAVL and HIQL, where the high-level value follows LAVL, and the low-level value and subgoal representation follow HIQL. Therefore, the high-level value is parameterized by $V^h(s, g) = -\|\varphi_S(s) - \varphi_G(g)\|_2$ and trained by (5.2). In contrast, the low-level value is defined by $V^l(s, g) = h(s, \phi(s, g))$ where $h$ is a standard MLP and $\phi(s, g)$ is a state-conditioned subgoal representation. Following HIQL, the low-level value is trained by the IVL loss:

$$\mathbb{E}_{(s,s')\in\mathcal{D},g\sim p^{\mathcal{D}}_{\text{mixed}}(\cdot|s)}[\ell_2^\kappa(r(s, g) + \gamma\tilde{V}^l(s', g) - V^l(s, g))].$$

One advantage of this implementation is that we can directly utilize the implementation of HIQL, including hyperparameters. However, as the low-level value accompanies the problems of HIQL, LAVL-HV shows lower performance compared to LAVL.

