# OpenReview forum: "Latent Representation Alignment for Offline Goal-Conditioned Reinforcement Learning"
_ICML.cc/2026/Conference — ICML 2026 regular_

### Official Review · Reviewer_HogB · 2026-03-09

**Soundness:** 3
**Presentation:** 3
**Significance:** 3
**Originality:** 3
**Overall Recommendation:** 3
**Confidence:** 4

**Summary:**

The paper studies offline goal-conditioned RL in long-horizon settings and argues that value overgeneralization is a key bottleneck. To address this, it proposes LAVL, which combines a latent-distance value parameterization, a local continuity regularizer, and hierarchical policy extraction adapted from prior work. The method is evaluated on OGBench and reports very strong gains over existing baselines, especially on long-horizon and stitching-heavy tasks.

**Compliance With Llm Reviewing Policy:**

Affirmed.

**Final Justification:**

While the paper is empirically promising, my overall assessment remains unchanged. In my view, the main idea is relatively simple: it adapts a dual-representation-style critic structure into the value function used for offline goal-conditioned RL and shows that this design can improve performance in some challenging settings. However, I still find the overall contribution somewhat limited. A substantial part of the paper’s intuition and structure appears to borrow from existing representation-based ideas, while the methodological novelty beyond this adaptation feels modest. Moreover, the paper is supported almost entirely through empirical results, without a comparably strong theoretical or mechanistic analysis to elevate the contribution beyond a useful architectural variant. As a result, although I acknowledge the practical gains, I do not find the work sufficiently strong or conceptually substantial for acceptance in its current form.

**Key Questions For Authors:**

1. Can you more cleanly disentangle the gains from the asymmetric latent-distance value form and the continuity regularizer?

2. Why was the specific asymmetric subtraction form $V(s,g)=-|\phi_s(s)-\phi_g(g)|$ chosen over other asymmetric designs such as bilinear, dot-product-based, or learned similarity functions?

3. Could the same local continuity regularizer also improve stronger baselines such as IQE or other value parameterizations?

4. What are the representational limitations, if any, of modeling value as a distance between separate state and goal embeddings?

5. How much of the final performance gain comes from the value design itself, versus the hierarchical policy extraction framework used on top of it?

**Limitations:**

No.The paper does not appear to include a dedicated limitations section. While some limitations are reflected indirectly in the experiments and analysis, the paper does not clearly state the scope and boundary conditions of the proposed design.

**Strengths And Weaknesses:**

The paper addresses an important problem in offline goal-conditioned RL, especially in long-horizon settings where value overgeneralization can hurt stitching and hierarchical policy extraction. A strength of the paper is that the proposed method is simple and intuitive: the latent-distance value parameterization is easy to understand, the continuity regularizer is lightweight, and the empirical evaluation on OGBench is fairly broad. The paper also presents a coherent high-level story connecting value geometry, local smoothness, and long-horizon performance, which makes the overall narrative easy to follow.

That said, I have several technical concerns. First, the method design is not sufficiently disentangled. LAVL combines multiple ingredients at once: an asymmetric latent-distance value form, a local continuity regularizer, and hierarchical policy extraction. While the overall method performs well, the paper does not clearly isolate which component is actually driving the gains. In particular, I did not find a clean ablation that separates the effect of the asymmetric latent-distance parameterization itself from the effect of the continuity regularizer.

Second, although the paper argues for the asymmetric value form $V(s,g) = -|\phi_s(s)-\phi_g(g)|$, it is not fully clear why this particular formulation is the right choice relative to other possible asymmetric designs. Since prior metric and quasimetric methods are discussed extensively, it would help to clarify whether the gains mainly come from asymmetry itself, from using separate encoders, or from relaxing stricter structural assumptions.

Third, the paper’s analysis remains somewhat limited relative to the strength of its conclusions. A central part of the paper is framed around overgeneralization and value geometry, but the supporting evidence is still mostly empirical and qualitative. In particular, it is not fully clear why the proposed latent-distance form should generalize equally well across both maze navigation and manipulation, especially since the gains on manipulation tasks appear more limited.

In terms of significance, I think the problem setting is important and the benchmark is meaningful. The proposed method is also reasonable and empirically promising. However, the current paper would be stronger with clearer component disentanglement and a more convincing explanation of why these specific design choices are responsible for the gains. Overall, I view this as a solid empirical paper with a sensible method, though its main claims are supported more strongly by empirical performance than by a sufficiently strong theoretical or mechanistic explanation.

---

> ### Author Rebuttal · Authors · 2026-03-31
>
> We appreciate your time and feedback on our paper and the opportunity to strengthen our results.
> We address your questions below.
>
> ---
> [Each table reports the average success rate over 5 datasets: {Point, Antmaze,Humanoidmaze}-giant-stitch, Cube-single-play, and Scene-play.]
>
> ### **Full results in the [link (Here)](https://drive.google.com/file/d/1wRwDVLurSEzl8iSx4KsfNCO-y8wLIC7h/view?usp=drive_link).**
>
> ---
> ## **Ablation Experiments on Value Architecture and Continuity Regularization**
>
> The performance gains of LAVL over the baseline stem primarily from the LAN-based value parameterization.
> Continuity regularization addresses residual local errors after LAN mitigates overgeneralization.
> To disentangle the contributions of LAN and continuity regularization, we compare four configurations that vary (i) the value parameterization and (ii) whether continuity regularization is used.
>
> ||HIQL|HIQL with Reg.|LAVL without Reg.|LAVL with Reg.|
> |-|-|-|-|-|
> |Average success rate|$18\pm2$|$19\pm2$|$65\pm2$|$80\pm2$|
>
> The largest improvement comes from replacing the MLP value function with LAN.
> Continuity regularization provides little benefit in HIQL but yields a further gain when combined with LAN.
> This suggests that LAN addresses the dominant bottleneck, while continuity regularization helps with remaining local errors (full results in **Table 11 at the link**).
>
> ---
> ## **Comparison with Alternative Asymmetric Value Architectures**
>
> We adopt LAN because it performed best in our preliminary experiments.
> To support this choice, we compare it with other asymmetric value parameterizations within LAVL:
> - Dot Product: $V(s,g)=\varphi_S(s)^T \varphi_G(g)$.
> - Bilinear: $V(s,g)=\varphi_S(s)^T W \varphi_G(g)$ where $W$ is a learnable matrix.
> - Learned Similarity: $V(s,g) = f([\varphi_S(s), \varphi_G(g)])$ where $f(\cdot)$ is a learnable MLP.
>
> ||LAN (ours)|Dot Product|Bilinear|Learned Similarity|
> |-|-|-|-|-|
> |Average success rate|$83\pm2$|$46\pm2$|$50\pm3$|$17\pm1$|
>
> **LAN consistently outperforms the alternative designs across datasets (Table 12 at the link)**.
> Dot-product and bilinear parameterizations remain competitive in some tasks but degrade in more challenging settings, while learned similarity performs poorly overall.
> This supports our claim that effective value learning depends not only on expressive power, but also on appropriate inductive bias.
>
> ---
> ## **Continuity Regularization for Baseline Value Architectures**
>
> To verify that continuity regularization also benefits other value architectures, we replaced LAN in LAVL with IQE, MRN, and Hilbert and evaluated the resulting variants under different regularization strengths (full results in **Table 13 at the link**).
>
> ||LAN (ours)|IQE|MRN|Hilbert|
> |-|-|-|-|-|
> |$w_c=0$|$65\pm2$|$32\pm4$|$34\pm4$|$45\pm2$|
> |$w_c=1$|$68\pm3$|$49\pm3$|$44\pm2$|$48\pm3$|
> |$w_c=10$|$76\pm3$|$50\pm2$|$43\pm2$|$50\pm2$|
>
> The table shows that continuity regularization improves average performance across architectures, although the gains are dataset-dependent, so we treat $w_c$ as a hyperparameter.
> LAN performs best at every value of $w_c$, indicating that its advantage is orthogonal to the effect of continuity regularization.
>
> ---
> ## **Discussion on Representational Limitation of LAN**
>
> Modeling the value as a distance between state and goal representations does not, in general, guarantee universal approximation over functions in $\mathcal{S}\times\mathcal{A}\to\mathbb{R}$.
> However, our empirical results suggest that this is not a major practical limitation.
> - Universal approximation is asymptotic and does not necessarily reflect the behavior of finite networks, and our goal is not to represent arbitrary functions but to learn value functions with low bias.
> - Although MLP value functions are universal approximators, our experiments show that they are more prone to overgeneralization; in GCRL, appropriate structural bias can instead promote more stable learning and more accurate generalization.
>
> ---
> ## **Ablation Experiments on Value Design and Hierarchy**
>
> To disentangle the contributions of value learning and hierarchy in LAVL, we ablate LAN-based value learning and the hierarchical policy.
> Removing LAN reduces LAVL to HIQL, replacing the hierarchical policy under LAN yields GCIVL+LAN, and removing both reduces to GCIVL.
> ||GCIVL|GCIVL+LAN|HIQL|LAVL (ours)|
> |-|-|-|-|-|
> |antmaze-medium-navigate|$72\pm8$|$88\pm1$|$96\pm1$|$97\pm1$|
> |antmaze-large-navigate|$16\pm5$|$52\pm3$|$91\pm2$|$92\pm1$|
> |antmaze-giant-navigate|$0\pm0$|$1\pm0$|$65\pm5$|$83\pm1$|
> |cube-single-play|$53\pm4$|$68\pm2$|$15\pm3$|$79\pm4$|
> |scene-play|$42\pm4$|$64\pm5$|$38\pm3$|$87\pm6$|
>
> - LAN improves performance under both flat and hierarchical policies, highlighting the importance of accurate value generalization in GCRL.
> - The hierarchy consistently helps with navigation, but on manipulation tasks, it only improves LAN-parameterized methods. This suggests that hierarchical policies are most effective when combined with LAN.

---

> > ### Author Rebuttal · Reviewer_HogB · 2026-04-02
> >
> > Thanks for the detailed rebuttal and the additional ablations. While these experiments improve empirical clarity, my overall assessment remains unchanged. I still think the work would benefit from substantially stronger theoretical or mechanistic analysis, since the main conclusions are still supported mostly empirically. Moreover, many of the central claims and intuitions seem quite close to prior dual-representation work; the main difference, as I understand it, is that this paper incorporates a related structure into the critic. As a result, the overall novelty and contribution still feel somewhat limited in my view. Therefore, I am inclined to maintain my original rating.

---

> > > ### Author Response · Authors · 2026-04-06
> > >
> > > We thank the reviewer for the follow-up comments. We respectfully clarify the issues that appear to underlie the remaining disagreement.
> > >
> > > ---
> > > ## **Evidence for the Main Claims**
> > > With all due respect, we disagree with the characterization that the paper’s conclusions are supported only "mostly empirically" in a weak sense. The claims of the paper are empirical/mechanistic claims about value parameterization and its interaction with hierarchy, and they are tested by controlled component-level evidence directed exactly at the core research questions.
> > >
> > > The paper does not rely on benchmark wins alone. We isolate the role of value architecture from that of the learning objective, compare LAN against alternative architectures in both flat and hierarchical settings, and analyze long-horizon and stitching behavior using quantitative measures such as success rate and order consistency. In our view, **this already constitutes a substantive mechanistic analysis of which component matters and in what regime, rather than a merely descriptive empirical narrative**.
> > >
> > > More importantly, **our claims are not theoretical statements** such as convergence rates or sample complexity. They are **empirically and scientifically testable claims** such as the effect of inductive bias in offline GCRL and the interaction between value parameterization and hierarchical policy. **Those claims are already directly examined by controlled ablations in the paper and in the rebuttal.**
> > >
> > > We also note that many outstanding papers in offline RL and GCRL are valued for identifying key bottlenecks and validating practical solutions **empirically rather than theoretically**. This is true of most of the major baselines discussed in our paper, including IQL [1], HIQL [2], and OTA [3], as well as metric-based representation methods [4,5,6] that inspire our design. **We therefore believe that the absence of additional formal theory should not by itself weaken a paper whose central claims are already systematically supported.**
> > >
> > > ---
> > > ## **Comparison with Prior Representation Learning Methods**
> > >
> > > We respectfully believe that the current assessment overstates the connection to prior work and, as a result, does not fully capture the distinction between our paper and the relevant literature. We hope the clarification below helps resolve this point.
> > >
> > > As discussed in Section 5, although the functional form of LAN is inspired by metric-based state representation methods [4,5,6], **the objective and central claim of our paper are fundamentally different.** Those works mainly focus on learning reusable representations, whereas our paper identifies value parameterization as a key bottleneck in offline GCRL and proposes LAN specifically as a value architecture tailored to that problem.
> > >
> > > The comment refers to "prior dual-representation work" but does not provide a specific citation. We therefore assume that it refers to "Dual Goal Representations" [7]. Beyond the broad similarity that both approaches involve state or goal representations, however, the core claims of that work and ours are fundamentally different.
> > >
> > > Like other representation learning approaches, [7] is primarily concerned with learning a reusable goal representation for downstream GCRL. By contrast, **our paper is not about learning a reusable representation.** Our central claim is that the inductive bias of the value function is a key factor in accurate generalization, and LAN is introduced specifically for this purpose. Moreover, [7] mainly focuses on an inner-product parameterization, while only secondarily considering a symmetric metric form in ablations. This inner-product parameterization is structurally very different from our LAN, and our experiments further show that this choice matters empirically.
> > >
> > > For these reasons, **we do not believe that [7] is the closest prior work to ours, nor that it is an appropriate basis for characterizing our central claims as substantially similar.**
> > >
> > > ---
> > > We therefore respectfully maintain that the paper’s contribution is supported by substantial quantitative evidence and is meaningfully distinct from the prior work discussed above. We appreciate the reviewer’s consideration.
> > >
> > > ---
> > > **References**
> > >
> > > [1] Kostrikov, Ilya, Ashvin Nair, and Sergey Levine. "Offline reinforcement learning with implicit q-learning."
> > >
> > > [2] Park, Seohong, et al. "Hiql: Offline goal-conditioned rl with latent states as actions."
> > >
> > > [3] Ahn, Hongjoon, et al. "Option-aware Temporally Abstracted Value for Offline Goal-Conditioned Reinforcement Learning."
> > >
> > > [4] Sermanet, Pierre, et al. "Time-contrastive networks: Self-supervised learning from video."
> > >
> > > [5] Ma, Yecheng Jason, et al. "VIP: Towards Universal Visual Reward and Representation via Value-Implicit Pre-Training."
> > >
> > > [6] Park, Seohong, Tobias Kreiman, and Sergey Levine. "Foundation policies with hilbert representations."
> > >
> > > [7] Park, Seohong, Deepinder Mann, and Sergey Levine. "Dual Goal Representations."

---

### Official Review · Reviewer_7zMc · 2026-03-11

**Soundness:** 3
**Presentation:** 2
**Significance:** 2
**Originality:** 3
**Overall Recommendation:** 4
**Confidence:** 4

**Summary:**

In the context of offline goal-conditioned reinforcement learning, the paper proposes Latent-Aligned Value Learning (LAVL), which consists of two components:

- a neural network called Latent Alignment Network (LAN) that represents value function as the negative Euclidean distance in embedding space between a state and a goal.
- a hierarchical structure with a high-level policy and a low-level policy where the subgoal embedding is not state dependent.

The proposition is evaluated on 22 datasets of OGBench and compared with several baselines.

**Compliance With Llm Reviewing Policy:**

Affirmed.

**Final Justification:**

I thank the authors for their replies. After this exchange, I'm not really convinced that claiming as contribution value overgeneralization or architectural change as primary cause is really well-justified, since value overgeneralization has to occur if the performance is bad and architectural change is a straightforward answer to this issue. For this reason, I believe that I correctly identified the main novel contributions of the paper in my original review.
However, the new experimental results with minimal tuning show a good performance of LAVL in the different environments. This addresses my concern about hyperparameter sensitivity. I would suggest the authors to present their experimental results with minimal tuning, since tuning in every environment, while providing higher perfomance, is not practical in my opinion.
Overall, my main concerns have been partly addressed. I have raised my score to weak accept.

**Key Questions For Authors:**

1. Did I miss anything regarding the design of LAVL (see my comment above)?
2. In Figure 3, does Hilbert corresponds to an architecture like LAN where the same encoder is used for states and goals?
3. Can you provide more explanation about the formula for the local continuity regularization term?
4. Is there any interpretation for the formula proposed for \delta?
5. is there a baseline that corresponds to LAVL where the same encoder is used for states and goals?
6. Would the findings in Section 4 really generalize to other methods/architectures?
7. Could you discuss what you did for hyperparameter tuning?
8. How did you select the tasks and datasets from OBGench? In particular, it's not clear to me that the proposed method would work well on a task like Puzzle.

**Limitations:**

Yes

**Strengths And Weaknesses:**

The proposed method, LAVL, combines existing techniques, while introducing with some key variations and achieves impressive performance on the selected 22 datasets. However, I think the design choices of the proposed method could be better justified and more details could be given for the experimental evaluation.

In particular, the experimental results should be provided with a discussion of hyperparameter tuning and explanation of how the tasks/datasets were selected. It seems that the SOTA results can only be achieved with hyperparameter values adapted to different tasks/datasets, which makes the proposition not so practical in my opinion. In addition, it seems that there are also slight differences in the training process for different tasks as well (Appendix C2).

The presentation is generally clear for someone familiar with this line of work, but it could be improved to be more accessible.

More detailed remarks:

The proposed method combines existing techniques with subtle variations. As far as I understand the main novel ideas are:

- Using distances in embedding space to constrain value function representation
- One embedding network for states and another for goals
- Local continuity regularization term for value learning

Those ideas are then implemented within existing methods (GCIVL and HIQL). Did I miss anything? I would suggest the authors to separate more clearly existing work with their propositions. For instance, Section 3 could recall GCIVL, HIQL and other relevant methods, while Section 4 could be dedicated to present LAVL and how it differs from previous work..
In addition, I would have appreciated more (e.g., empirical or theoretical) justifications for these design choices:

- In Figure 3, does Hilbert corresponds to an architecture like LAN where the same encoder is used for states and goals?
- Can you provide more explanation about the formula for the local continuity regularization term? It seems to me that the regularization term proposed by Giammarino et al, written with finite differences would lead to another formula.
- Is there any interpretation for the formula proposed for \delta?
- In the experiments, is there a baseline that corrresponds to LAVL where the same encoder is used for states and goals?

Regarding experiment results, I like Section 4. However, since the experiments are specifically performed with GCIVL and QRL. Would the findings really generalize to other methods/architectures?
Regarding the SOTA results, could you discuss what you did for hyperparameter tuning?

The exposition could be better organized and the writing could be more polished:

- The paper tends to discuss the experimental results too early in my opinion (e.g., paragraph in lines 83-92, col. 1 page 2 or paragraph in lines 264-271, col. 1, page 5). For a reader who reads the paper linearly, this is not ideal since the reader doesn't have all the elements yet to truly understand and appreciate those plots.
- The paper should be more self-contained and relevant concepts should be recalled or presented the first time they appear in the text (e.g., quasimetric, QRL objective, or goal sampling distributions…).
- There is no conclusion

Minor points:

- The authors should be more precise: they evaluated on 22 datasets, not 22 tasks.
- Appendix C2: quasimetric architects -> quasimetric architectures?
- Appendix C2: which is two orders -> which is one order?

---

> ### Author Rebuttal · Authors · 2026-03-31
>
> We appreciate your time to review our paper and your feedback. We address your questions below.
>
> ### **Full results in the [link (Here)](https://drive.google.com/file/d/1wRwDVLurSEzl8iSx4KsfNCO-y8wLIC7h/view?usp=drive_link).**
>
> ---
> ## **Our Contribution**
>
> Our contribution is broader than combining existing ingredients.
> We **identify value overgeneralization as a central bottleneck in offline GCRL, isolate the role of value parameterization in causing this failure mode, and propose a value architecture that directly addresses it**.
> This diagnosis-and-remedy viewpoint is the core of the paper and is also how the paper is framed in the abstract and controlled analysis.
> LAN is introduced as a structural inductive bias for goal-conditioned value learning, and when combined with hierarchical policy extraction, it yields strong empirical gains.
>
> ---
> ## **On Hilbert Value Parameterization**
>
> In Figure 3, Hilbert is the shared-encoder variant of LAN.
> While prior Hilbert (metric) parameterizations were introduced to learn reusable state representations for downstream tasks, we use LAN to induce accurate value generalization.
> In this sense, our contribution lies not only in the architecture itself, but also in demonstrating its effectiveness in GCRL.
> Importantly, the shared-encoder baseline of LAVL is already included in Figure 6, where LAN consistently outperforms Hilbert.
> The advantage of LAN over other architectures is further demonstrated in Figure 3 and **Table 8 at the link**.
>
> ---
> ## **Continuity Regularization**
>
> Our regularizer is not intended as the exact finite-difference form of [1].
> Both methods aim to stabilize the local geometry of the value function, but [1] regularizes the gradient norm, whose finite-difference form would be slope-based and depend on the distance between neighboring states.
> In contrast, our regularizer uses a hinge function to penalize value variation between sampled transitions.
>
> The formula for $\delta$ is motivated by the Bellman equation.
> Given $(s,s')$ and $g$, we expect at convergence that $V(s,g)-V(s',g)\approx r(s,g)-(1-\gamma)V(s',g)$.
> This suggests that $|V(s,g)-V(s',g)|$ is upper bounded by $1+(1-\gamma)|V(s',g)|$, so $\delta$ can be interpreted as a rough upper bound on the value difference.
>
> ---
> ## **Generalization of Our Findings**
>
> On the architecture side, our conclusion is already supported by evidence beyond IQE-specific results.
> Figure 12 shows that architectures with inductive bias reduce value overgeneralization.
> Figures 3 and 6 further show that LAN improves over MLP, IQE, MRN, and Hilbert, with both flat and hierarchical policies.
>
> On the method side, Sections 4 and 5 show that QRL’s advantage on navigation tasks stems primarily from IQE rather than from the objective itself.
> To examine whether this finding extends beyond GCIVL, we implemented GCXVL, a goal-conditioned extension of XQL [2], using MLP, IQE, and LAN.
>
> **Figure 13 at the link** visualizes the learned value functions.
> For GCXVL with MLP, overgeneralization appears in multiple regions, and the local value geometry is unstable.
> By contrast, GCXVL with IQE and LAN both show accurate generalization, emphasizing the role of structural inductive bias.
> ||GCXVL+MLP|GCXVL+IQE|GCXVL+LAN|
> |-|-|-|-|
> |antmaze-medium-navigate|$85\pm2$|$90\pm1$|$89\pm2$|
> |antmaze-large-navigate|$41\pm1$|$44\pm9$|$54\pm2$|
> |cube-single-play|$47\pm4$|$28\pm4$|$54\pm4$|
> |cube-double-play|$37\pm6$|$40\pm3$|$51\pm1$|
>
> Moreover, the above table (details in **Table 9 at the link**) shows that GCXVL with LAN tends to match or outperform GCXVL with MLP and GCXVL with IQE.
> Together, these results indicate that the findings of Section 4 extend beyond GCIVL to GCXVL.
>
> ---
> ## **Dataset Selection and Hyperparameter Tuning**
>
> Following prior OGBench works such as HIQL, OTA, and CGCIVL, we evaluate LAVL on maze navigation and robotic manipulation.
> In addition, we evaluate LAVL on puzzle datasets under the HIQL settings.
> ||LAVL|CGCIVL|OTA|HIQL|
> |-|-|-|-|-|
> |puzzle-3x3-play|$16\pm1$|$11\pm2$|$2\pm1$|$12\pm2$|
> |puzzle-4x4-play|$4\pm2$|$3\pm2$|$1\pm1$|$0\pm0$|
>
> Most hyperparameters were inherited from HIQL, and our search was systematic: we tuned $\gamma\in\\{0.99,0.998,0.999\\}, \kappa\in\\{0.7,0.9\\}$, and $\beta^l\in\\{3,10\\}$ for each environment (maze, cube, scene), $w_c\in\\{0,1,10\\}$ for each task family ({point, ant, humanoid}maze, cube, scene), and $\beta^h\in\\{0.5,1,2\\}$ for each dataset.
> A reduced-tuning variant with $(\beta^h,\beta^l)=(1,3)$ fixed across all datasets shows only modest degradation in **Table 10 at the link**, suggesting that LAVL does not rely on heavy dataset-specific tuning.
>
> Finally, we thank the reviewer for the helpful suggestions on paper organization and will incorporate them in the revision.
>
> ---
> **References**
>
> [1] Vittorio and Qureshi. "Goal Reaching with Eikonal-Constrained Hierarchical Quasimetric Reinforcement Learning."
>
> [2] Garg, Divyansh, et al. "Extreme Q-Learning: MaxEnt RL without Entropy."

---

> > ### Author Rebuttal · Reviewer_7zMc · 2026-04-02
> >
> > I'd like to thank the authors for their clarifications and for providing additional experimental results. I believe these explanations should be added in the final version of the paper.
> >
> > I have a couple of follow-up questions:
> > - Regarding the stated first contribution, isn't value overgeneralization a general well-known phenomenon in offline RL? In addition, isn't it a straightforward reason for any low performance in deep RL?
> > - Regarding hyperparameter tuning, even for the reduced-tuning variant, \gamma and \kappa have to be specifically tuned for each task. Is that correct?

---

> > > ### Author Response · Authors · 2026-04-06
> > >
> > > We are grateful for the constructive discussion, and we will incorporate your suggestions into the final version of the paper to further strengthen the exposition.
> > > We are happy to address the follow-up questions below. With these clarifications, we hope that our core contributions are recognized.
> > >
> > > ---
> > > ## **On Value Overgeneralization as a Contribution**
> > >
> > > While poor generalization may be cited as a high-level explanation for low performance in deep RL, our contribution is not merely to invoke this intuition, but to **systematically diagnose the primary cause, and show that a concrete architectural change can resolve it.** This distinction becomes clearer when viewed in the context of recent prior works:
> > >
> > > - [1] reports a phenomenon closely aligned with our notion of overgeneralization, where value estimates propagate based on Euclidean proximity rather than environment topology. While their online GCRL experiments demonstrate that a deep network alleviates this issue, **their solution does not generalize to offline settings, and even degrade performance** (Figure 18 in [1]).
> > > In contrast, our architectural solution clearly addresses value overgeneralization in offline GCRL, despite using a shallow network of depth 3.
> > > - Eik-HIQL [2] introduces a regularizer that encourages value functions to follow environment geometry. While this improves alignment, Table 6 shows that Eik-HIQL exhibits poor performance in long-horizon tasks and stitching datasets. The empirical result suggests that the **issue is only partially addressed with Eik-HIQL**.
> > > - OTA [3] analyzes value learning through order consistency, identifying that value estimates become non-monotonic along optimal trajectories in long-horizon settings. This is an insightful proxy for value quality, and we include order consistency analysis in **Appendix B.3**. However, [3] focuses on trajectory-wise monotonicity and **does not directly analyze broader value generalization across the state space**.
> > >
> > > Taken together, while prior works have observed related phenomena or proposed partial remedies, this is the first work to directly attribute the failure to value parameterization and validate this claim through controlled experiments at scale.
> > >
> > > ---
> > > ## **Further Clarification on Hyperparameters**
> > >
> > > $\gamma$ and $\kappa$ do not HAVE to be separately tuned for each task in order for LAVL to perform well. Our additional experiments show that LAVL remains strong even when these hyperparameters are fixed more broadly, indicating substantial robustness in practice. Per-task tuning is only used to further optimize performance (note here even this seperate tuning is for different task, not for different datatsets), and this is standard for essentially all offline GCRL methods rather than being specific to ours.
> > > Below, we provide the detailed explanation.
> > >
> > > ### (1) Robustness on $\kappa$.
> > > In OGBench's HIQL implementation, $\kappa=0.7$ is the default choice. LAVL already performs well with this default value; however, we found that $\kappa=0.9$ provides additional gains in maze tasks. To assess the sensitivity, we conducted additional experiments using fixed $\kappa=0.7$ and $(\beta^h,\beta^l)=(1,3)$ across all datasets. **In Table 15 at the [link (Here)](https://drive.google.com/file/d/18wKtIlUoimOxVQbXd_Y8mOULEgZ-WZqn/view?usp=drive_link)**, minimally tuned LAVL shows only minor performance degradation, and this suggests that $\kappa$ does not require precise per-task tuning in practice.
> > >
> > > ### (2) Discount factor ($\gamma$) plays a fundamentally different role.
> > > In all GCRL methods, $\gamma$ **directly determines the effective planning horizon**, thus it is inherently task-dependent. Tasks such as giant mazes require $\gamma$ close to 1 to propagate sparse rewards over thousands of steps. Using a smaller $\gamma$ leads to limited reward propagation, which cannot be corrected by learning dynamics. Therefore, selecting $\gamma$ based on the task horizon is standard practice in prior methods.
> > >
> > > Importantly, our LAN architecture provides a clear advantage in this regard. As shown in the order consistency analysis in Appendix B.3, when $\gamma$ is close to 1, **MLP-parameterized value functions become unstable and often fail to learn**. In such cases, reward propagation becomes ineffective for long-horizon tasks, which is a key limitation of prior methods.
> > >
> > > In contrast, LAN exhibits **stable convergence and high order consistency over a wide range of $\gamma$ values**. This allows us to reliably set $\gamma$ in long-horizon tasks, where proper reward propagation is essential.
> > >
> > > ---
> > > **References**
> > >
> > > [1] Wang, Kevin, et al. "1000 layer networks for self-supervised rl: Scaling depth can enable new goal-reaching capabilities."
> > >
> > > [2] Giammarino, Vittorio, Ruiqi Ni, and Ahmed H. Qureshi. "Physics-informed Value Learner for Offline Goal-Conditioned Reinforcement Learning."
> > >
> > > [3] Ahn, Hongjoon, et al. "Option-aware temporally abstracted value for offline goal-conditioned reinforcement learning."

---

### Official Review · Reviewer_K1sf · 2026-03-12

**Soundness:** 3
**Presentation:** 4
**Significance:** 3
**Originality:** 2
**Overall Recommendation:** 5
**Confidence:** 3

**Summary:**

This paper proposes Latent-Aligned Value Learning (LAVL), which integrates the Latent Alignment Network (LAN) into GCIVL and extracts a policy through hierarchical AWR (similar to HIQL). There are two key motivations. First, the authors observe that a value network without structural inductive bias, e.g., MLP, suffers from value overgeneralization. It describes the phenomenon where states close in Euclidean space but distant temporally can be erroneously assigned high value estimates. Second, compared with the QRL value objective, TD learning is necessary to obtain good performance on manipulation tasks. Based on these motivations, LAN employs two distinct encoders for states and goals to enforce asymmetry bias in the case of value overgeneralization, and then LAVL combines LAN and GCIVL with an additional local continuity regularization. Empirical evaluation across 22 OGBench tasks shows performance superiority against other GCRL baselines, which indicates the benefit from latent alignment in LAN.

**Compliance With Llm Reviewing Policy:**

Affirmed.

**Final Justification:**

As highlighted in the Strengths section of my review, I appreciate that LAVL, as a simple method, effectively improves performance over recent baseline methods on an unsaturated benchmark and provides meaningful insights into the important problem of offline value learning.

During the rebuttal, the authors have provided additional experiments addressing my concerns about CRL value leakage. Preliminary results on "value overgeneralization" of broader offline GCRL algorithm families beyond value-based approaches have better connected this work to the literature. Furthermore, efforts have been made towards robustness to hyperparameter tuning. Finally, although it is no longer my main concern, the resemblance to some prior works may limit the work's novelty, as also noted by other reviewers.

In summary, I am therefore satisfied with the empirical results of this paper and remain positive about the manuscript, recommending its acceptance.

**Key Questions For Authors:**

See weaknesses.

**Limitations:**

See weaknesses. And the authors are encouraged to add sections discussing limitations and conclusions.

**Strengths And Weaknesses:**

***Strengths:***

1. This paper is well-organized and easy to follow. Math notations are well-defined and consistent. Important takeaways are provided at the end of the section. Experimental results are well-summarized into tables and figures. I did not encounter difficulty in understanding the paper’s content.
2. The proposed method is well motivated by the observed value overgeneralization. Section 4 approaches the bottleneck of GCRL through analyzing two design choices: value network architecture and value learning objective. The first axis is instantiated as MLP and IQE, and the second axis is instantiated as GCIVL and QRL. This controlled analysis contributes positively to the paper.
3. LAN is simple and intuitive. It works well on considered tasks and outperforms considered baselines.
4. This work engages closely with the recent GCRL literature. The bottleneck of offline value learning is an important research question in general. LAVL’s policy design also echoes recent research advocating horizon reduction [1].
---
***Weaknesses:***

1. The paper’s novelty is limited by the following:
    1. Value overgeneralization has been observed in the literature, e.g., Figure 9 in [2].
    2. The proposed LAN architecture closely resembles the value network architecture commonly used in Contrastive RL [2, 3]. Usually, states and goals are encoded by distinct networks in CRL, and thus, the asymmetric latent representation is not a new idea.
    3. Other key design choices, including the IVL objective, continuity regularization, and hierarchical AWR policy extraction, are the same or only incremental to prior work.

    I am aware that findings in the online setting may not naturally transfer to the offline setting. However, could the authors further distinguish LAN from the CRL architectural design, considering their similarity? If necessary, the authors could augment the two design axes in Section 4 by adding the CRL value objective or design a fair comparison with CRL.

2. Based on my understanding, $\mathcal{L}\_{reg}$ intuitively encourages states close in Euclidean space to have similar embeddings $\phi_{\mathcal{S}}$ (and thus to have similar values). According to Table 4, the continuity regularization seems to help except in Humanoidmaze and Cube. And LAVL does not beat OTA in Humanoidmaze-navigate. Considering these observations, are there any deeper reasons and insights into why Humanoidmaze causes difficulty for LAVL? Is it possible that the continuity regularization leads to state embedding collapse/ambiguity, given the high-dimensionality of Humanoid? The authors are encouraged to dive deeper to explain why LAVL works well and may fail (limitations).
---
***References:***

[1] Horizon Reduction Makes RL Scalable (2025)

[2] 1000 Layer Networks for Self-Supervised RL: Scaling Depth Can Enable New Goal-Reaching Capabilities (2025)

[3] Demystifying the Mechanisms Behind Emergent Exploration in Goal-conditioned RL (2025)

---

> ### Author Rebuttal · Authors · 2026-03-31
>
> Thank you for your positive evaluation of our paper and your insightful feedback. We address your questions below.
>
> ---
> ## **Value Overgeneralization**
>
> Our claim is not merely that value overgeneralization exists, but that **its resolution depends on the inductive bias** of the value architecture rather than on network depth scaling.
> As the reviewer noted, **online GCRL** results shown Figure 9 in [1] show that a shallow network (depth 4) assigns high values based on Euclidean proximity despite the maze wall, whereas a deep network (depth 64) alleviates this issue.
> However, **offline GCRL** results shown Figure 18 in [1] show that, in two out of three environments, performance declined drastically as depth scaled from 4 to 64.
> Moreover, Appendix A.4 in [1] shows that quasimetric architectures such as MRN do not scale consistently across environments.
> Taken together, these observations strengthen our claim that LAN is a key component in addressing value overgeneralization, despite using a shallow network of depth 3.
>
> ---
> ## **Comparison with CRL**
>
> While CRL also encodes states and goals in a latent space, it differs from our approach in both functional form and learning objective:
>
> - **Functional Form:** We directly parameterize the goal-conditioned value function as $V(s,g)=-||\varphi_S(s)-\varphi_G(g)||$. In contrast, CRL parameterizes an **unnormalized log Q-function** as $\log Q(s,a,g)=-||\phi(s,a)-\varphi(g)||+C(g)$, where $C(g)$ is an unknown goal-dependent constant (see Lemma 4.1 in [2]). Thus, CRL does not directly parameterize the value function in the way LAN does.
> - **Value Learning Objective:** LAN is designed for stable TD learning. By contrast, CRL’s parameterization is tied to a contrastive learning objective. Because CRL uses the InfoNCE objective, it effectively parameterizes a **similarity function between $(s,a)$ and $g$, rather than a goal-conditioned value function.**
>
> We appreciate the reviewer for noting the similarity.
> However, given the differences discussed above, we do not believe that CRL’s parameterization admits a principled direct comparison with TD learning or quasimetric learning as in Section 4.
>
> ---
> ## **Discussion on Novelty**
>
> While LAVL uses standard offline GCRL building blocks such as expectile regression and hierarchical policy extraction, **the paper’s novelty does not lie in introducing yet another objective variant**.
> The core contribution is to **identify value overgeneralization as a fundamental failure mode in offline GCRL, show through controlled analysis that this failure is tightly linked to value parameterization, and introduce LAN as a structural remedy tailored to this problem**.
>
> In this sense, our contribution is not a superficial combination of existing components, but a new **diagnosis-and-design principle** for offline goal-conditioned value learning: the paper isolates the bottleneck, demonstrates why standard parameterizations fail, and proposes LAN that mitigates this failure and integrates naturally with practical hierarchical policy frameworks.
> To the best of our knowledge, this is the first work to make this connection explicit in offline GCRL and to validate it empirically at this scale.
>
> ---
> ## **Discussion on Continuity Regularization**
>
> The main issue is not representation collapse per se; rather, the regularizer introduces additional bias that is beneficial only when local value noise is a dominant error source.
>
> $L_{Reg}$ does not encourage Euclidean-nearby states to share the same embedding; rather, it penalizes excessive changes in $V(s,g)-V(s',g)$ over observed transitions $(s,s')$ across goals $g$.
> When local value errors remain significant, it can improve stability, but when such local irregularities are not the main bottleneck, it may provide limited benefit or even introduce bias.
> This explains why it is more clearly helpful in PointMaze-like settings, where we observed locally "wrinkled" value surfaces.
>
> For Humanoidmaze, the dominant challenge appears to be long-horizon and high-dimension, where an additional regularization may be counterproductive.
> For Cube, whose horizon is much shorter than that of the other tasks, reward propagates more smoothly, and local fluctuation is less central.
>
> ---
> ## **Discussion on OTA**
>
> For the result where OTA performs slightly better than LAVL on Humanoidmaze-navigate, one plausible explanation is that the horizon-reduction mechanism of OTA is especially effective in navigation datasets with long trajectories.
> In contrast, on stitch datasets, which consist of shorter trajectory segments, OTA receives less benefit from horizon reduction, whereas LAVL is less constrained by this dataset limitation and therefore achieves stronger performance.
>
> ---
> **References**
>
> [1] Wang, Kevin, et al. "1000 layer networks for self-supervised rl: Scaling depth can enable new goal-reaching capabilities."
>
> [2] Eysenbach, Benjamin, et al. "Contrastive learning as goal-conditioned reinforcement learning."

---

> > ### Author Rebuttal · Reviewer_K1sf · 2026-04-03
> >
> > I confirm that I have carefully read the rebuttal and other reviews. I thank the authors for their response to my comments. The clarification has addressed my concerns. Below are some comments:
> >
> > 1. As highlighted in the Strengths section of my review, I appreciate that LAVL, as a simple method, effectively improves performance over recent baseline methods on an unsaturated benchmark and provides meaningful insights into the important problem of offline value learning. I am therefore satisfied with the empirical results of this paper and remain positive about the manuscript, recommending its acceptance.
> >
> > 2. In terms of CRL, the authors' responses all make sense, and I buy them. However, in my original review, I intended to suggest that it would further strengthen the evidence for "value overgeneralization" if some visualization of the similarity function in CRL, analogous to that presented in the 1000-layer paper, were included in the study. As noted by the authors, the 1000-layer paper's central conclusion does not hold for offline CRL, which raises the expectation that (Q-)value leakage may still occur, similar to what is shown in your Figure 2. Essentially, they all describe the same pathology where states close in Euclidean space but distant temporally tend to have similar value. Furthermore, without a more thorough examination of major offline GCRL algorithm families beyond value-based approaches—such as CRL and flow-based RL—I would not go that far to claim: “value overgeneralization is a fundamental failure mode in offline GCRL.”
> >
> > 3. Although these are not my primary concerns, after reading other reviews, I agree that hyperparameter tuning and the resemblance to some prior works may limit the broader impact of this paper on the community.
> >
> > For these reasons, I maintain my current rating but remain positive about this work.
> > ___
> > Update: Thanks for your serious consideration of my further comment. My concerns have been fully resolved. I also see additional efforts on testing robustness to hyperparameter tuning. I decide to increase my score to 5.

---

> > > ### Author Response · Authors · 2026-04-06
> > >
> > > We sincerely thank the reviewer for the positive feedback and continued support. We are happy to provide our responses to the last comments.
> > >
> > > ---
> > > ## **Analysis on CRL Value**
> > >
> > > We thank the reviewer for the insightful suggestion. Following this feedback, we conducted additional experiments visualizing the value landscape learned by CRL, analogous to Figure 2. Since CRL admits multiple choices of similarity function, we considered both the **dot-product similarity** used in the original CRL paper [1] and the **L2-distance-based similarity** used in follow-up work such as [2].
> > >
> > > In addition, we implemented two further offline GCRL baselines for broader comparison: **GCFQL**, a goal-conditioned extension of FQL that uses a flow-matching policy within an actor-critic framework, and **GCXVL**, a goal-conditioned variant of XQL that learns a goal-conditioned value function via Gumbel regression for MaxEnt RL.
> > >
> > > **Figure 15 in the [link (Here)](https://drive.google.com/file/d/18wKtIlUoimOxVQbXd_Y8mOULEgZ-WZqn/view?usp=drive_link)** visualizes the learned value functions of these methods (and the learned similarity function in the case of CRL). The results suggest that generalization issue arises across multiple offline GCRL algorithms:
> > >
> > > - CRL exhibits irregular errors throughout the state space. Its learned landscape contains substantial local fluctuations (“ripples”), indicating that the learned similarity function does not form a smooth geometry aligned with the environment dynamics. This appears to stem from the contrastive nature of the CRL objective. These results suggest that, while overgeneralization is a prominent and recurring failure mode, it is not the only way in which learned goal-conditioned geometry can become misaligned with temporal structure.
> > > - With dot-product similarity, CRL clearly exhibits overgeneralization: states that are close to the goal in Euclidean distance receive high similarity even when they are far in temporal distance. In contrast, with L2-based similarity, CRL shows little visible overgeneralization apart from the irregular local errors described above.
> > > - GCFQL and GCXVL show a pattern consistent with GCIVL. MLP-parameterized value functions exhibit overgeneralization, whereas LAN yields much more accurate generalization. This provides additional evidence that the inductive bias of LAN can be beneficial beyond GCIVL, and may extend to other offline GCRL algorithms as well.
> > >
> > > Overall, these additional results strengthen our diagnosis that **reliable offline GCRL depends critically on accurate generalization that is aligned with temporal distance**.
> > >
> > > ---
> > > We appreciate your feedback and will revise the paper to include these additional value visualizations, broaden the discussion of failure modes beyond overgeneralization.
> > > We sincerely appreciate the time and effort you devoted to the review process.
> > >
> > > ---
> > > **References**
> > >
> > > [1] Eysenbach, Benjamin, et al. "Contrastive learning as goal-conditioned reinforcement learning."
> > >
> > > [2] Wang, Kevin, et al. "1000 layer networks for self-supervised rl: Scaling depth can enable new goal-reaching capabilities."

---

### Official Review · Reviewer_wYAe · 2026-03-13

**Soundness:** 4
**Presentation:** 4
**Significance:** 4
**Originality:** 3
**Overall Recommendation:** 4
**Confidence:** 3

**Summary:**

This paper presents an important improvement over offline goal conditioned RL. The authors demonstrate the utility of correct inductive biases in value estimation and propose Latent Value Learning (LAVL) that integrates latent representation and hierarchical planning. The method is successfully validated on OGBench and shows promising improvements in the frontiers of the offline RL desiderata: trajectory stitching and long-horizon planning.

**Compliance With Llm Reviewing Policy:**

Affirmed.

**Key Questions For Authors:**

Q1. Are there any particular patterns in results or expected success rate that emerge when testing across datasets?

Q2. How are the task specific hyperparameters obtained (crucially, how are the two new temperature betas tuned and how sensitive are they)? Can you do a more transparent analysis like the online bandit procedure suggested in [3] . With that analysis and the weaknesses above addressed, I am willing to increase my score.

[3] Jackson, Matthew Thomas, et al. "A clean slate for offline reinforcement learning." arXiv preprint arXiv:2504.11453 (2025).

**Limitations:**

yes

**Strengths And Weaknesses:**

**Strengths**:

- Approach performs strongly on an unsaturated benchmark

- Trajectory stitching and long horizon solutions

- Well-written and easy to follow

- Clear results

**Weaknesses**:

- Missing the discussion of relevant work in hierarchical rl and horizon reduction [1] [2]

- Unclear how the maze visualization is color-coded. The overgeneralization is difficult to assess just by looking at those Figures

---

> ### Author Rebuttal · Authors · 2026-03-31
>
> We thank the reviewer for the careful reading and positive feedback. We address the questions and concerns below.
>
> ---
> ### **Discussion on Hierarchical RL and Horizon Reduction**
>
> [Note: As the references were listed only as [1] and [2] without explicit titles/authors, we inferred the intended works from the context of the reviewer’s comment.]
> Regarding hierarchical RL, our related work section already discusses several closely related methods, including HIQL, CGCIVL, OTA, and Eik-HIQL.
> If the reviewer clarifies which specific works were intended, we would be happy to provide a more detailed discussion.
>
> For horizon reduction, we assume the reviewer is referring to "Horizon Reduction Makes RL Scalable" and we discuss its relationship to LAVL on that basis.
> That work shows that reducing the effective horizon, through $n$-step returns, hierarchical policies, or both, can substantially improve scalability in challenging long-horizon tasks.
> LAVL is aligned with this perspective, since its hierarchical policy design also reflects the logic of horizon reduction.
> However, our work focuses on a different bottleneck in offline goal-conditioned value learning.
> Our findings suggest that addressing value overgeneralization requires an appropriate structural inductive bias, and that achieving robust performance across diverse domains depends on a stable TD learning objective.
> Thus, while both works address the challenge of long-horizon offline GCRL, LAVL does so by emphasizing value representation and stable value learning rather than by focusing on horizon reduction.
>
> ---
> ### **Improving the Clarity of Maze Visualizations**
>
> We will refine the color scheme and explicitly indicate regions of overgeneralization to facilitate more accurate interpretation of the goal-conditioned value function.
> However, in Appendix B, we already included two additional analyses to provide a more comprehensive evaluation of overgeneralization.
>
> * Kendall order consistency (Appendix B.3): We introduce Kendall order consistency as a quantitative proxy for value quality, assessing whether the learned values are appropriately ordered along optimal trajectories. Using optimal-trajectory datasets from OTA, we demonstrate that the MLP network exhibits a marked decline in consistency as the horizon increases, while LAN maintains greater reliability.
> * Enhanced Visualization (Appendix B.4): We added a legend to make the value scale explicit and included visualizations across multiple evaluation goals, making overgeneralization easier to identify than with standard heatmaps.
>
> ---
> ### **Result Patterns and Success Rates**
>
> In Section 6, LAVL demonstrates substantially greater stability as task difficulty increases, particularly in long-horizon scenarios and trajectory stitching datasets.
> Specifically, while baseline methods experience severe performance degradation as the maze scale increases from medium to giant, LAVL maintains consistent stability.
> For example, LAVL exhibits a marginal relative drop of only 9.6%, compared to a 23.1% drop for OTA and a 58.3% drop for CGCIVL.
> This stability is consistently observed in the stitch datasets as well.
> In stitch datasets, successful performance requires composing multiple segments to propagate reward signals across the entire horizon, given the short trajectories.
> Consequently, all baseline methods exhibit substantial degradation when trajectory stitching is required.
> In contrast, LAVL exhibits a minimal performance drop, demonstrating that its advantages are pronounced when robust long-horizon value propagation and stitching are essential.
>
> ---
> ### **Hyperparameter Selection and Sensitivity Analysis**
>
> Most hyperparameters were inherited from HIQL, and others were selected in a principled manner:
> - $\gamma\in\\{0.99,0.998,0.999\\}, \kappa\in\\{0.7,0.9\\}$, and $\beta^l\in\\{3,10\\}$ were tuned for each environment (maze, cube, scene).
> - $w_c\in\\{0,1,10\\}$ was tuned for each task ({point, ant, humanoid}maze, cube, scene).
> - Only $\beta^h\in\\{0.5,1,2\\}$ was tuned per dataset.
>
> Importantly, **LAVL is robust to the AWR temperature betas.** LAVL with fixed betas, where $(\beta^h,\beta^l)=(1,3)$, shows only minimal degradation in **Table 9 at the [link](https://drive.google.com/file/d/1wRwDVLurSEzl8iSx4KsfNCO-y8wLIC7h/view?usp=drive_link)**. This suggests that LAVL does not rely on heavy dataset-specific tuning.
>
> Our hyperparameter selection relies on online evaluation.
> However, the online bandit procedure in "A Clean Slate for Offline Reinforcement Learning" considers a more specific problem, selecting among many candidate policies under a limited online evaluation budget.
> By contrast, our setting compares a small number of hyperparameters through standard evaluation runs, rather than relying on a dedicated bandit-based strategy.
> While a wide search could yield even stronger performance, we believe it is a notable strength of LAVL that it already surpasses prior work with moderate tuning.

---

> > ### Author Rebuttal · Reviewer_wYAe · 2026-04-03
> >
> > Apologies for the missing citations.
> >
> > [1] Park, Seohong, et al. "Transitive RL: Value Learning via Divide and Conquer." arXiv preprint arXiv:2510.22512 (2025).
> >
> > [2] Park, Seohong, et al. "Horizon reduction makes rl scalable." arXiv preprint arXiv:2506.04168 (2025).
> >
> > Thank you for the rebuttal. The point of the evaluation with the bandit is not to solve a more specific problem. On the contrary, it is the most general setting for an offline RL method and I cannot increase my score in light of such a fundamental misunderstanding. [...] "moderate'' tuning and arguments regarding an arbitrary assessment of a "small number of hyperparameters" are not enough to substantiate the robustness of the algorithms in the offline rl sense.

---

> > > ### Author Response · Authors · 2026-04-06
> > >
> > > We thank the reviewer for the clarification and the continued support.
> > > In our earlier response, we focused on explaining how hyperparameters were selected.
> > > We now understand that the main point of the suggested bandit evaluation is to assess robustness under a broader offline RL protocol.
> > > Addressing your feedback, we further conducted the additional analysis below.
> > >
> > > ---
> > > ## **Evaluation Procedure**
> > >
> > > We first collect a pool of online evaluation scores from policies trained by LAVL and HIQL.
> > > Concretely, we train **24 candidate policies for each method** from fixed hyperparameter ranges and random seeds.
> > >
> > > - **LAVL**: we vary the high-level temperature $\beta^{h}\in\\{0.5, 1, 2, 3\\}$ and the continuity regularization coefficient $w_{c}\in\\{0, 1, 10\\}$, and train 2 random seeds for each configuration.
> > > - **HIQL**: we vary $\beta^{h}\in\\{0.5, 1, 2, 3\\}$ and train 6 random seeds per value.
> > >
> > > For other parameters, such as the goal-sampling ratio and the low-level temperature, we use the same values for both methods.
> > > Each policy is evaluated online, and the resulting scores are used for the subsequent bandit evaluation.
> > >
> > > Following the spirit of the bandit-based evaluation in Section 4 from [1], we perform policy selection over a fixed pool of candidate policies under different tuning budgets.
> > > Since our experiments are conducted on OGBench, where performance is summarized by success rate rather than episodic return, we use one policy-level score per candidate policy, obtained from **50 evaluation episodes and averaged over 5 goals**.
> > > Specifically, we subsample **8 candidate policies** and run a **UCB bandit** over them.
> > > We evaluate performance as a function of the tuning budget, **up to a maximum of 200 policy evaluations**.
> > > For each budget, results are averaged over **500 repeated bandit rollouts**, and confidence intervals are estimated using **2000 bootstrap samples**.
> > >
> > > To keep the environment selection consistent, we evaluate on **6 datasets**: antmaze-giant-navigate-v0, antmaze-giant-stitch-v0, humanoidmaze-large-navigate-v0, and humanoidmaze-large-stitch-v0 from the maze-navigation domain, and cube-single-play-v0 and scene-play-v0 from the robotic-manipulation domain.
> > >
> > > ---
> > > ### **Results:**
> > >
> > > **In Figure 14 at the [link (Here)](https://drive.google.com/file/d/18wKtIlUoimOxVQbXd_Y8mOULEgZ-WZqn/view?usp=drive_link)**, we present the bandit-based evaluation of HIQL and LAVL on four OGBench datasets, where the y-axis denotes mean success rate and the x-axis denotes the number of policy evaluations.
> > > Generally, a method performs better if its curve is closer to the top-left corner of the plot, indicating higher success rates with fewer online policy evaluations.
> > > The right side of the curve corresponds to performance under increasingly large tuning budgets.
> > >
> > > Across all 6 datasets, **LAVL consistently outperforms HIQL over the full range of tuning budgets**.
> > > The advantage is especially pronounced on the stitch tasks, where HIQL remains far below LAVL even with larger policy-evaluation budgets.
> > > This suggests that LAVL provides a stronger and more robust candidate policy pool, rather than benefiting only from favorable policy selection after extensive tuning.
> > > Importantly, the ordering between the two methods is stable across the budget range, suggesting that **LAVL’s improvement is not driven merely by aggressive tuning, but by better policy quality and robustness**.
> > >
> > > ---
> > > ## **Comparison with TRL**
> > >
> > > While both LAVL and TRL [2] study offline GCRL in long-horizon settings, they address fundamentally different bottlenecks and make different technical contributions.
> > > TRL attributes the main challenge to bias accumulation in TD learning and proposes a divide-and-conquer value update rule based on the triangle inequality.
> > > In contrast, our work identifies value overgeneralization as the key bottleneck, where values generalize according to Euclidean proximity rather than true temporal reachability.
> > > Accordingly, our contribution is not a new backup rule, but a new value archiecture with appropriate inductive bias, namely LAN, together with continuity regularization and hierarchical policy learning.
> > > Put differently, TRL mainly changes how values are backed up across horizon, whereas our work mainly changes how values are represented so that they generalize reliably.
> > > Therefore, although the two works share a long-horizon offline GCRL setting, their novelty claims and technical contributions are clearly different.
> > >
> > > ---
> > > We will therefore revise the paper to include these additional hyperparameter tuning results and to broaden the discussion of related work.
> > > We sincerely appreciate the time and effort you devoted to the review process.
> > >
> > > ___
> > > **References**
> > >
> > > [1] Jackson, Matthew Thomas, et al. "A clean slate for offline reinforcement learning." arXiv preprint arXiv:2504.11453 (2025).
> > >
> > > [2] Park, Seohong, et al. "Transitive RL: Value Learning via Divide and Conquer." arXiv preprint arXiv:2510.22512 (2025).

---

### Decision · Program_Chairs · 2026-04-30

**Decision:**

Accept (regular)

**Comment:**

This submission studies long-horizon value learning can fail due to erroneous generalization of goal-conditioned value functions. The authors propose Latent-Aligned Value Learning (LAVL), centered on a distance-based value parameterization with separate state/goal encoders, combined with a continuity regularizer and hierarchical policy extraction.

Empirically, LAVL is very strong on OGBench (22 datasets), achieving best performance on 20/22, with especially notable gains on long-horizon navigation and trajectory-stitching settings where several baselines degrade substantially.

The reviewer pool leans positive (two weak accepts, one accept, one weak reject). The main concerns were (i) novelty / disentanglement of components and (ii) hyperparameter robustness / evaluation protocol. during rebuttal, the author provide  additional ablations showing the LAN critic is the primary driver of gains (with the regularizer providing a secondary improvement), comparisons against alternative asymmetric similarity parameterizations.

Overall, the method is simple and conceptually not very novel. However, the insightful practical analysis, ablations and the strong performance justify the acceptance.